# Identification of a highly conserved neutralizing epitope within the RBD region of diverse SARS-CoV-2 variants

Yanqun Wang [1,2,3,11], An Yan[4,11], Deyong Song [5,11], Maoqin Duan[6,11], Chuangchuang Dong[5], Jiantao Chen[1], Zihe Jiang[4], Yuanzhu Gao[4], Muding Rao[5], Jianxia Feng[5], Zhaoyong Zhang[1], Ruxi Qi[4], Xiaomin Ma [4], Hong Liu[5], Beibei Yu[5], Qiaoping Wang[5], Mengqi Zong[5], Jie Jiao[5], Pingping Xing[5], Rongrong Pan[5], Dan Li[5], Juxue Xiao[1], Junbo Sun[5], Ying Li[5], Linfeng Zhang[5], Zhenduo Shen[5], Baiping Sun[5], Yanyan Zhao[5], Lu Zhang[7], Jun Dai[7], Jingxian Zhao [1], Lan Wang [6] ✉, Changlin Dou[5] ✉, Zheng Liu [4] ✉ & Jincun Zhao [1,8,9,10] ✉

The constant emergence of SARS-CoV-2 variants continues to impair the efficacy of existing neutralizing antibodies, especially XBB.1.5 and EG.5, which showed exceptional immune evasion properties. Here, we identify a highly conserved neutralizing epitope targeted by a broad-spectrum neutralizing antibody BA7535, which demonstrates high neutralization potency against not only previous variants, such as Alpha, Beta, Gamma, Delta and Omicron BA.1-BA.5, but also more recently emerged Omicron subvariants, including BF.7, CH.1.1, XBB.1, XBB.1.5, XBB.1.9.1, EG.5. Structural analysis of the Omicron Spike trimer with BA7535-Fab using cryo-EM indicates that BA7535 recognizes a highly conserved cryptic receptor-binding domain (RBD) epitope, avoiding most of the mutational hot spots in RBD. Furthermore, structural simulation based on the interaction of BA7535-Fab/RBD complexes dissects the broadly neutralizing effect of BA7535 against latest variants. Therapeutic and prophylactic treatment with BA7535 alone or in combination with BA7208 protected female mice from the circulating Omicron BA.5 and XBB.1 variant infection, suggesting the highly conserved neutralizing epitope serves as a potential target for developing highly potent therapeutic antibodies and vaccines.

Latest estimates from the Director-General of the World Health Organization show that the full death toll associated with the COVID-19 pandemic was more than 20 million as of November 2023 (https://covid19.who.int/). These sobering data not only point to the impact of the pandemic but also to the need for effective prophylactic and therapeutic drugs. Multiple mAbs, either used alone or in combination, have been granted Emergency Use Authorization (EUA) or approval for therapeutic or preventive use during the COVID-19 pandemic[1–7].

Combination use or cocktail of therapeutic antibodies usually provides greater resistance to emerging variants and a lower risk of the arise of virus escape mutants during antibody therapeutic than a single-antibody treatment[8,9].

As the SARS-Cov-2 continues to evolve, most neutralizing antibodies have lost their activity[1–6,10–12]. LY-CoV1404 is the only authorized antibody therapy that could cover current sub-lineages BA.1, BA.2, BA.4, and BA.5 of the Omicron variant[13,14]. However, recently diverse

new Omicron subvariants continue to emerge intensively under the pressure of humoral immunity established through vaccination and infection, including BA.2.38.1, BA.2.74, BA.2.75, BA.2.76, BA.2.77, BA.2.79, BA.2.80, BA.4.6, BQ.1, BQ.1.1, XBB.1, XBB.1.5, XBB.1.16, XBB.1.9, XBB.2.3, CH.1.1 and EG.5[14–24]. These new subvariants further impair the efficacy of the available antibody therapy and humoral immunity established through vaccination and infection[14,25]. Till now, clinicians have no authorized therapeutic mAb for treatment. It raises a huge concern, especially for immunocompromised patients who do not adequately respond to COVID-19 vaccines. Neutralizing antibodies targeting the Spike RBD are usually categorized into four or seven classes based on blocking the Spike binding to ACE2 receptor and recognition of the 'up' or 'down' state of the three RBDs in one Spike[26,27]. Multiple key interacting residues or hotpot mutations in RBD, such as G339D, R346T, S371L, K417N, N440K, K444M, G446S, E484A, F486V, Q493R, G496S, and Q498R[28,29], resulted from adaptive mutations, conferred escape to most of the neutralizing antibodies. Most human monoclonal antibodies reported to date have not demonstrated true potent broad-spectrum efficacy against previous and emerging variants. The evolving Omicron subvariants illustrate the necessity of the discovery of novel conservative antigenic epitopes, which are valuable for developing broad-spectrum and potent neutralizing antibodies as well as designing more effective vaccines against future variants.

Here, we identified a conserved neutralizing epitope present on the RBD region targeted by a highly potent neutralizing mAb BA7535. In vitro, neutralizing potency of BA7535 used alone or in combination with previously reported BA7208[30] against over 30 previous and recently emerged SARS-CoV-2 variants performed well. Cryo-electron microscopy analysis elucidates the broad-spectrum mechanism of BA7535 and reveals the conserved antigenic epitope in SARS-CoV-2 RBD. This conserved cryptic RBD epitope and potent neutralizing capacity of BA7535 indicate its potential therapeutic application for SARS-CoV-2.

## Results

### Numerous Omicrons subvariants continue to emerge
Diverse new SARS-CoV-2 variants are constantly emerging. Key substitutions compared to wild-type stains in the receptor-binding domain (RBD) in the previous and recent emerging variants are shown in Fig. 1a. A number of new Omicron subvariants related to BA.4/5 and BA.2.75 have emerged and shown remarkable antibody evasion capacities, in particular BQ.1, BQ.1.1, BF.7, XBB, XBB.1.16, XBB.1.9, and CH.1.1. The new mutations in the recent emerging Omicron subvariants that differ from previous variants are marked in blue, including G339H, E340K, R346T/S, K356R, L368I, K444N/T, V445P, N450D, L452M, N460K, T478R, F486V/S/P, F490V and S494P. These subvariants show strong patterns of convergent evolution at the antigenic sites of the S receptor-binding domain (RBD), notably 346, 452, 460, and 486. While BQ.1 and BQ.1.1 have gained the special K444T mutation, a distinct F490S mutation is found in XBB. The new mutations bring more possibilities for the Omicron variants to escape antibody therapy. Striking antibody evasion by emerging SARS-CoV-2 Omicron subvariants drives the development of conserved epitopes and broadly neutralizing antibodies (bNAbs).

### Identification of a highly potent neutralizing antibody BA7535
To overcome the persistent immune escape and identify conserved epitopes of emerging Omicron subvariants, potential anti-SARS-CoV-2 antibodies were identified through immunizing human antibody transgenic mice BA-huMab followed by phage display[31] (Supplementary Fig. 1). The transgenic mice could generate fully human antibodies without the need of complex humanization process commonly used in murine antibodies. In detail, six transgenic mice were immunized and then boosted with RBD proteins of the SARS-CoV-2 BA.1 variant. Serum

from immunized mice collected 4 days after the last boost was used for titer determination by enzyme-linked immunosorbent assay (ELISA), and a large number of mAbs binding to Omicron RBD are produced in mice (Supplementary Fig. 2). Spleens of each immunized mouse were subjected to phage library construction followed by capturing with BA.1 RBD protein. Positive Single-chain variable fragments (ScFvs) detected by ELISA were picked out for sequencing. Multiple ScFvs with unique sequences showed their ability to block ACE2 receptor-binding to RBD and were converted into IgG1 antibodies for further in vitro evaluation. Among them, six mAbs, including BA7535, BA7525, BA7523, BA7501, BA7533, and BA7530, demonstrated broadly blocking activity against RBDs of 9 representative SARS-CoV-2 variants with BA7535 exhibiting the strongest blocking potency (Supplementary Fig. 3).

### In vitro neutralizing potency of BA7535 against emerging SARS-CoV-2 variants
Next, we evaluated the in vitro neutralizing potential of BA7535 used alone or in combination with BA7208 with antibody LY-CoV1404 (also known as Bebtelovimab, produced in-house) as reference. BA7208 is another anti-SARS-CoV-2 antibody demonstrating neutralizing potency against multiple variants as reported previously[30]. The neutralization potency and breadth of each antibody were examined against over 30 previous and recently emerged variants in human huh-7 cells using VSV pseudovirus system decorated with the respective spike protein (Fig. 1b and Supplementary Fig. 4). Combined with mutation sites of these variants in Fig. 1a for analysis, BA7208 was vulnerable to R346 mutations and impaired by BA.4.6/BF.7/BA.4.6.1 (R346T) and XBB (R346T). LY-CoV1404 was evaded by BA.2.38.1 (K444N), BQ.1 (K444T), BQ.1.1(K444T) and XBB(V445P + G446S) due to K444N/T mutations or the combination of V445P/G446S[7]. Importantly, BA7535 alone could neutralize all tested previous and recently emerged SARS-CoV-2 variants. D614G, Alpha, BQ.1.1, CH.1.1, and XBB only impaired the potency of BA7535, but none of the variants can escape its neutralization. In detail, BA7535 exhibits potent neutralizing efficacy to the Omicron variant BA.1-BA.5 ($IC_{50}$ = 1.7-40.2 ng/mL) as well as BQ.1, BQ.1.1, CH.1.1 and XBB subvariants ($IC_{50}$ = 42.8–274.4 ng/mL). The combination of BA7535 and BA7208 provided broader neutralization potency and higher resistance to immune evasion by SARS-CoV-2 variants than alone (Fig. 1b).

We next evaluated the neutralization potency of BA7535 using authentic SARS-CoV-2. BA7535 potently neutralized authentic SARS-CoV-2 virus BA.1, BA.2, BA.5, XBB.1, XBB.1.5, XBB.1.9.1 and EG.5 with $IC_{50}$ of 0.38 ng/mL, 1 ng/mL, 2 ng/mL, 47 ng/mL, 63 ng/mL, 34 ng/mL and 41 ng/mL, respectively (Fig. 2a, b). The binding affinity of BA7535 to the BA.1 RBD protein compared to BA7208 was examined by Surface Plasmon Resonance (SPR) (Fig. 2c). BA7535 showed a higher affinity to BA.1 RBD than BA7208 with the measured equilibrium constant (KD) as $0.10 \pm 0.02$ nM and $1.81 \pm 0.26$ nM, respectively. To explore the potential of their combination use, a Bio-Layer Interferometry (BLI) based competitive binding assay was performed, which indicated that they are non-competing antibodies, thus providing the potential for combined application in the treatment of COVID-19 (Supplementary Fig. 5).

### ADCC and ADCP activity
The interaction with Fc receptors can lead to killing of virus-infected cells through a variety of immune effector mechanisms was also evaluated, including Antibody-Dependent Cellular Cytotoxicity (ADCC) and Antibody-Dependent Cellular Phagocytosis (ADCP). The ADCC activity against CHO-K1 cells expressing spike of SARS-CoV-2 reference strain was characterized using a reporter bioassay (Promega, G7940) with Jurkat cells as effector cells and luciferase substrate as reacting regent for monitoring Fc receptor signaling (Fig. 2d). BA7208 demonstrated the strongest ability to mediate ADCC with $IC_{50}$ values as 5.17 ng/mL, but BA7535 almost ca not induce ADCC. LY-CoV1404

 

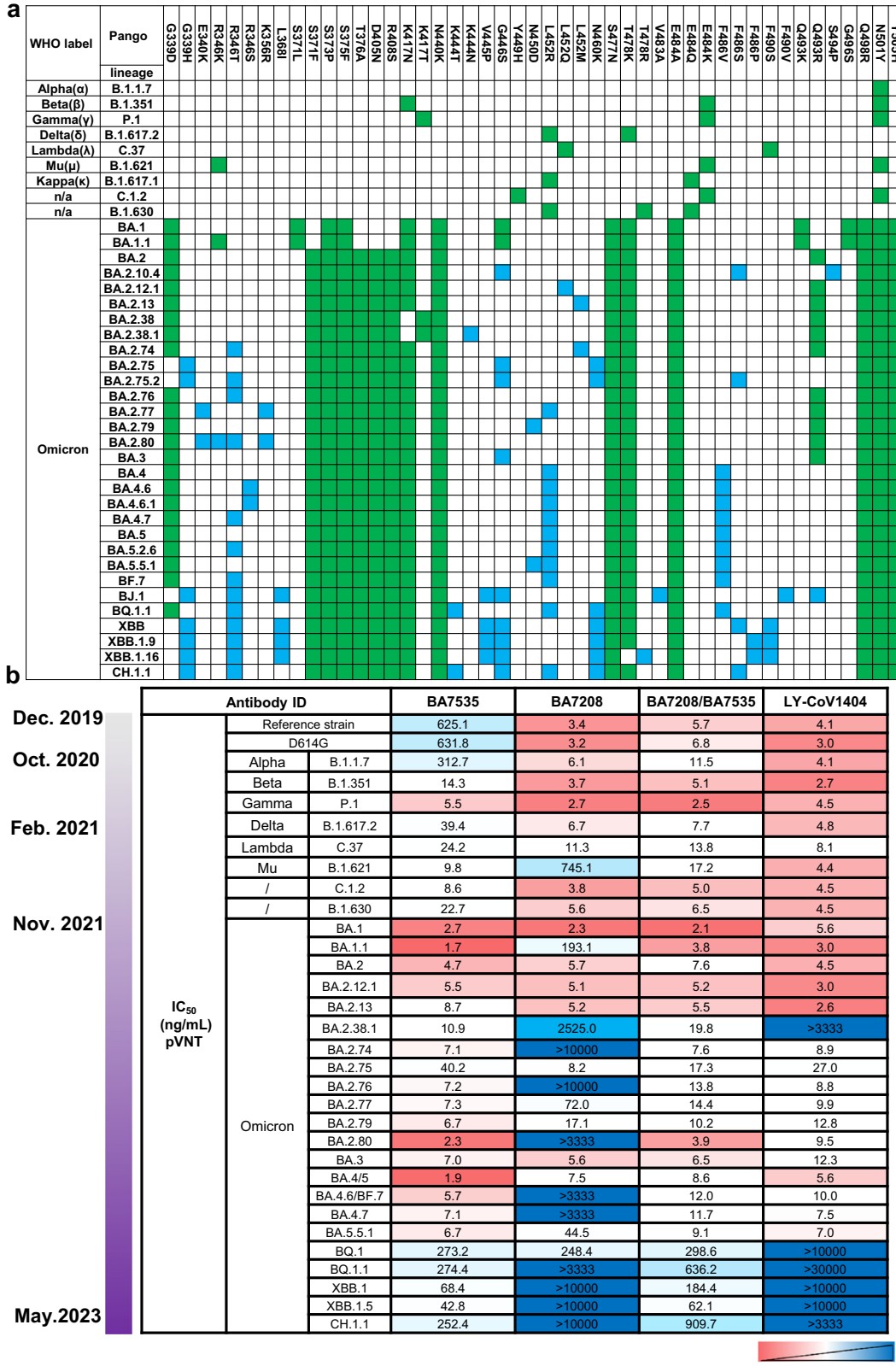

**Fig. 1 | Broadly neutralizing antibodies against previous and recently emerged SARS-CoV-2 variants. a** Key substitution in RBD in previous and current variants compared with the parental variant. The new mutation sites in Omicron subvariants that differ from BA.2 are marked in blue. **b** Neutralization activity of BA7535, BA7208, their combination, and LY-COV1404 against over 30 previous and recently emerged variants were examined using a pseudovirus system, and IC$_{50}$ values were shown. Data were collected from two biological replicates and represented as Mean ± SD.

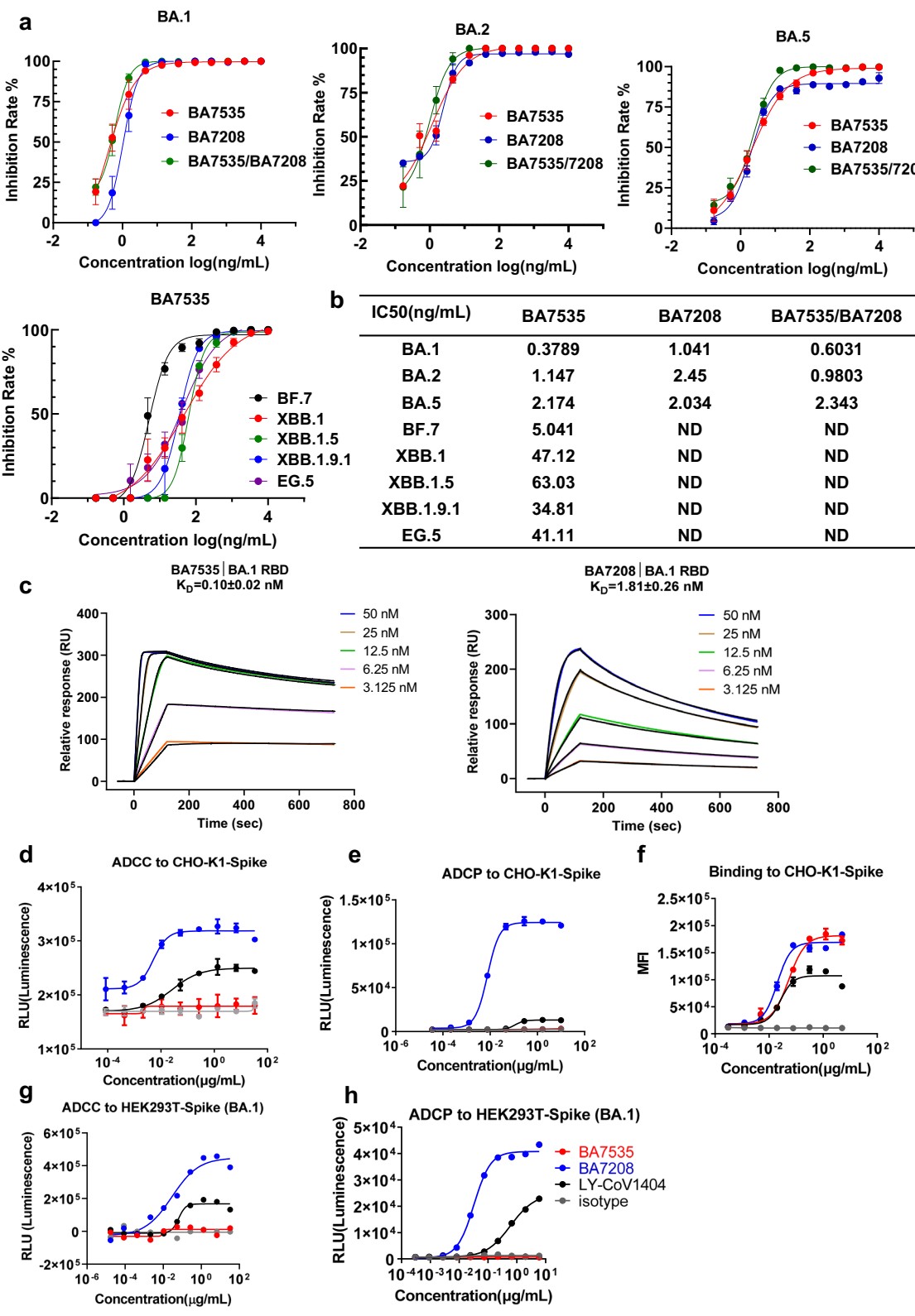

**Fig. 2 | Neutralizing activity, affinity, and antibody effector functions analyses of BA7535 and BA7208. a**, **b** Neutralization curves against the authentic SARS-CoV-2 variants (BA.1, BA.2, BA.5, BF.7, XBB.1, XBB.1.5, XBB.1.9.1 and EG.5) are shown for each of the indicated antibodies. Three biological replicates were performed. ND not detected. The average ± SD from two independent experiments with technical duplicates is shown. **c** KD values of BA7535 and BA7208 to BA.1 RBD examined by SPR. Data were collected from two biological replicates and represented as Mean ± SD. **d**–**h** ADCC and ADCP of BA7535, BA7208, and LY-CoVl404. Experiments were performed in duplicate. Three biological replicates were performed. Data were presented as Mean ± SEM. **d** ADCC activity of mAbs with CHO-K1 cells expressing SARS-CoV-2 spike (reference strain) as target cells. **e** ADCP activity of mAbs with CHO-K1 cells expressing SARS-CoV-2 spike (reference strain) as target cells. **f** Cell binding ability of mAbs against CHO-K1 cells expressing SARS-CoV-2 spike (reference strain). **g** ADCC activity of mAbs with HEK293T cells expressing SARS-CoV-2 BA.1 spike as target cells. **h** ADCP activity of mAbs with HEK293T cells expressing SARS-CoV-2 BA.1 spike as target cells. Irrelevant mAb with the same constant region was used as an isotype. Source data are provided as a Source Data file.

exhibited moderate ability to mediate ADCC. Similar to the ADCC results, BA7208 displayed a superior ability to mediate ADCP in an ADCP assay using Jurkat-FcγRIIA-H131 cells (Vazyme) as effector cells, with $IC_{50}$ values as 7.72 ng/mL (Fig. 2e). BA7535 and LY-CoV1404 showed no or little ability to mediate ADCP in this assay. To explore whether BA7535's weak binding to the spike of reference strain contributes to its weak effector function, cell-based binding was detected, and BA7535 demonstrated similar binding ability compared to BA7208 (Fig. 2f). Similar results were observed with HEK293T cells expressing SARS-CoV-2 BA.1 spike as target cells (Fig. 2g, h). In brief, BA7208 has the ability to mediate ADCC and ADCP, but BA7535 cannot.

### Cryo-electron microscopy analysis of the SARS-CoV-2 Spike trimer in complex with BA7535-Fab

To further understand the structural basis for its broad-spectrum properties and the neutralizing mechanism of BA7535, we determined the structure of the stable Omicron BA.2 Spike trimer in complex with BA7535-Fab using Cryo-EM. Details of sample preparation, data

collection, and cryo-EM analysis are present in the Methods section, Supplementary Table 1 and Supplementary Fig. 6. The structure of the Spike protein in association with BA7535-Fab was refined to 2.4 Å overall resolution. However, local resolution in the RBD/Fab sub-complex is only 4 Å, further local refinement focused in the sub-complex was performed, and resolution was improved to 3.07 Å. In this sub-map, analyses of the interactions between RBD and Fab were analyzed more accurately. Figure 3a shows an Omicron BA.2 Spike binding with three BA7535-Fabs on the top of the RBDs in the Spike, and all three RBDs are in the 'up' conformation.

Despite the significant antigenic shifts and structural changes of Omicron RBD, BA7535 still binds to the Omicron BA.2 RBD at an interface covering an area of 722 Å². The epitope in the Omicron RBD for BA7535 is composed of 7 interacting residues, including T415, D420, Y421, A475, N487, Y489 and R493. Figure 3b reveals six hydrogen bonds and one salt bridge formed between residues from BA7535-Fab and residues from Omicron BA.2 RBD (Supplementary Table 2). The hydrogen bonds include T415-Y106, D420-Y106, Y421-L103, A475-

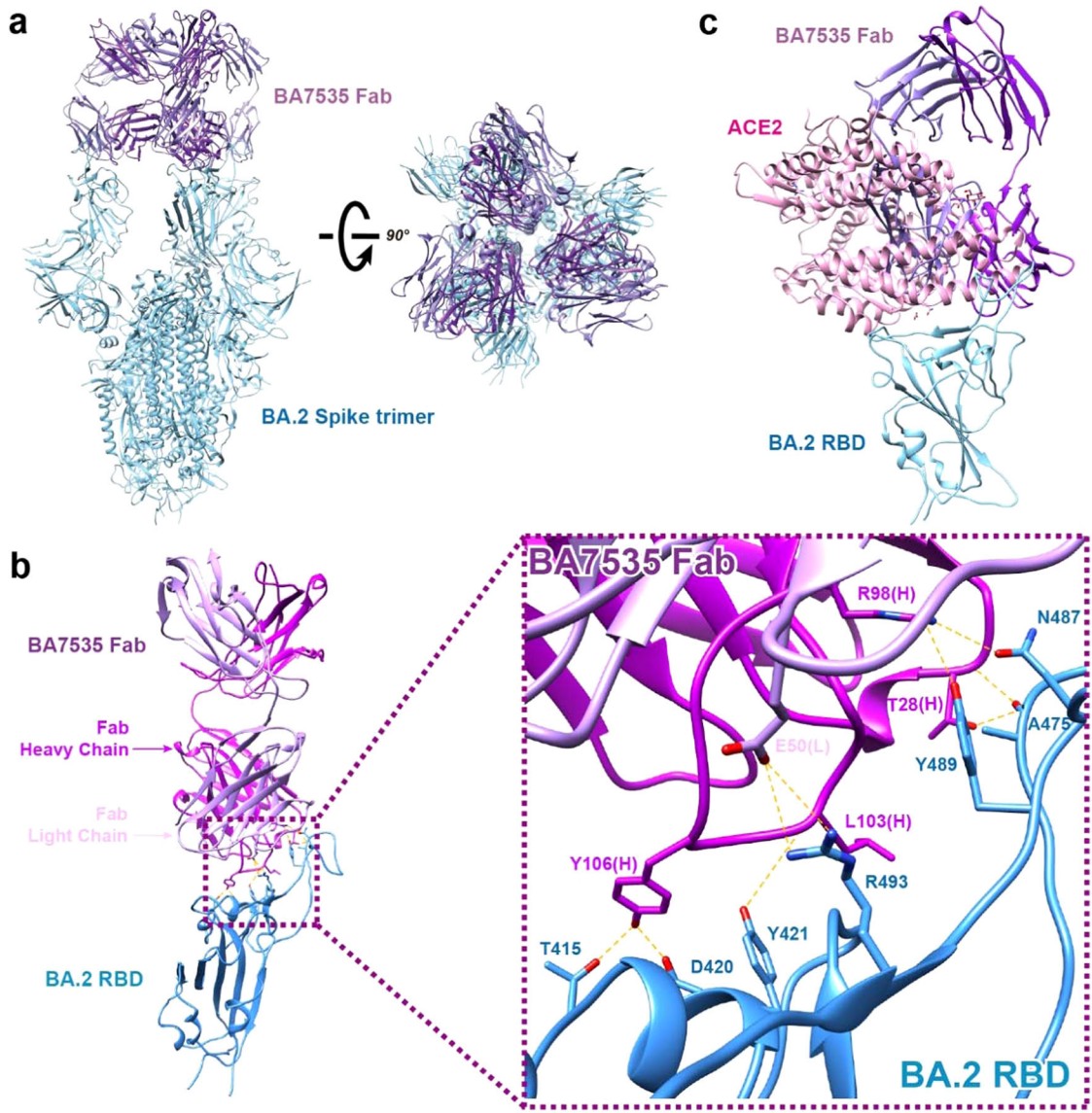

**Fig. 3 | Cryo-electron microscopy of Omicron BA.2 Spike protein with BA7535-Fab. a** The complex of three BA7535-Fabs (dark purple, Fab heavy chain; and light purple, Fab light chain) with Omicron BA.2 Spike Trimer (cyan). **b** The complex of one BA7535-Fab with Omicron BA.2 Spike RBD, zoomed-in views of BA7535-Fab binding site on Omicron BA.2 RBD, side chain of residues that forms the hydrogen bonds and salt bridge are displayed. H, heavy chain; L, light chain. **c** The complex of BA.2 RBD and BA7535-Fab is superimposed with complex of RBD and ACE2 (pink).

T28, N487-R98, and Y489-R98 (fore residues are from Omicron BA.2 RBD and hind residues are from BA7535-Fab) and the salt bridge is formed between R493-E50. To investigate the mechanism of antibody BA7535 neutralization, we superimposed the Spike RBD/ACE2 complex (PDB ID: 6VW1) with the solved structure of the completed RBD/BA7535-Fab. Figure 3c shows BA7535 partially overlaps with the ACE2 epitope, elucidating that the BA7535-Fab, by binding to the RBD of the Spike, occupies the binding site of the RBD to ACE2, which in turn directly blocks the binding of ACE2 to the RBD and thus blocking the virus invasion into host cells.

Next, we compared the binding site of BA7535 with antibodies that are licensed for clinical use, including LY-Cov555 (PDB ID: 7KMG)[1], LY-Cov016 (PDB ID; 7C01)[2], A23-58.1 (PDB ID: 7LRS)[32], REGN10933 (PDB ID: 6XDG)[3], 2196 (PDB ID: 8D8Q)[33], SA55 (PDB ID: 7Y0W)[34], 2130 (PDB ID: 8D8Q)[33], LY-Cov1404 (PDB ID: 7MMO)[7], REGN10987 (PDB ID: 6XDG)[3], Vir-7831 (PDB ID: 7R6X)[35], S309 (PDB ID: 7TLY)[36], SA58 (PDB ID: 7Y0W)[34], and our previously reported antibody BA7208 (PDB ID: 7XDB)[30] (Supplementary Fig. 7). Among these antibodies, 2130, LY-Cov1404, REGN10987, VIR-7831, S309, SA58, and BA7208 bind to the side portion on RBD, which is outside the ACE2 binding site and recognize both 'up' and 'down' RBDs (Supplementary Fig. 7h–n), according to the classification criteria[26,35,37], these antibodies belong to Class 3. In contrast, antibodies BA7535, LY-Cov555, LY-Cov016, A23-58.1, REGN10933, 2196, and SA55, bind to the top portion of RBD that block the ACE2 binding (Supplementary Fig. 7a–g), and bind only to 'up' RBDs (Supplementary Fig. 8), these antibodies are categorized as Class 1.

To further compare the similarities and differences between BA7535-Fab and these licensed antibodies, we use PISA to analyze the interface interaction between antibodies and Omicron BA.2 RBD (Supplementary Table 3). In those antibodies, the overall solvation free energies of RBD/LY-Cov555, RBD/2196, RBD/SA55, RBD/2130, RBD/LY-Cov1404, RBD/REGN10987, RBD/Vir-7831, RBD/S309 and RBD/SA58 are lower than RBD/BA7535-Fab, while RBD/LY-Cov016, A23-58.1, REGN10933, and BA7208 are higher (Supplementary Table 3). It seems that RBD/LY-Cov555, RBD/2196, RBD/SA55, RBD/2130, RBD/LY-Cov1404, RBD/REGN10987, RBD/Vir-7831, RBD/S309 and RBD/SA58 have stronger affinity than BA7535. However, when we compared the binding site of these antibodies and ACE2, we found that only BA7535, LY-Cov555, LY-Cov016, A23-58.1, REGN10933, 2196, and SA55 occupy the ACE2 binding region on the RBD effectively (Supplementary Fig. 7a–h).

In addition to the antibodies that are licensed for clinical use, there are several reports that characterized the latest-developed antibodies binding to spike or RBD by cryo-EM and X-ray crystallography[38–41]. we also compared the binding site of BA7535 with these antibodies, and Supplementary Fig. 9 illustrated that most newly developed antibody epitopes are located on the top of RBD, similar with BA7535, blocking the ACE2 binding site. Among these antibodies, BA7535, Omi-3, Omi-2, Omi-12, Omi-25, and BA.2-36 have their epitopes that directly overlap with ACE2 binding site, and more important, BA7535 and Omi-25 share unique contact residues on the top of the RBD. Based on the structural characteristics, BA7535-Fab binds to the top portion, thus avoids most of the mutational sites in RBD (Fig. 4), it reduces the risk of being affected by most mutations in spike protein and enhances the broad spectrum. We also download 221 human SARS-CoV-2 RBD-specific mAbs with available structures from the PDB (https://www.rcsb.org/) (Supplementary Table 4). Epitope residues and buried surface area (BSA) of the 221 RBD-specific mAbs as well as BA7535 were determined (Supplementary Fig. 10) using the PDBePISA server (https://www.ebi.ac.uk/msd-srv/prot_int/). Comparative analysis of epitopes of BA7535 and 221 published RBD-specific mAbs indicated that most of the previously reported neutralizing antibodies target one or two major antigenic sites, single or double mutations of concern at key positions always lead to immune escape, while the epitope residues and buried surface area (BSA) for each epitope residue of BA7535 are more dispersive and appear more resistant to most mutants. In vitro neutralizing potency experiments also confirmed the broad spectrum of BA7535 (Fig. 1a). In conclusion, among these antibodies mentioned above, BA7535 is one of the most efficient with broad neutralization potency.

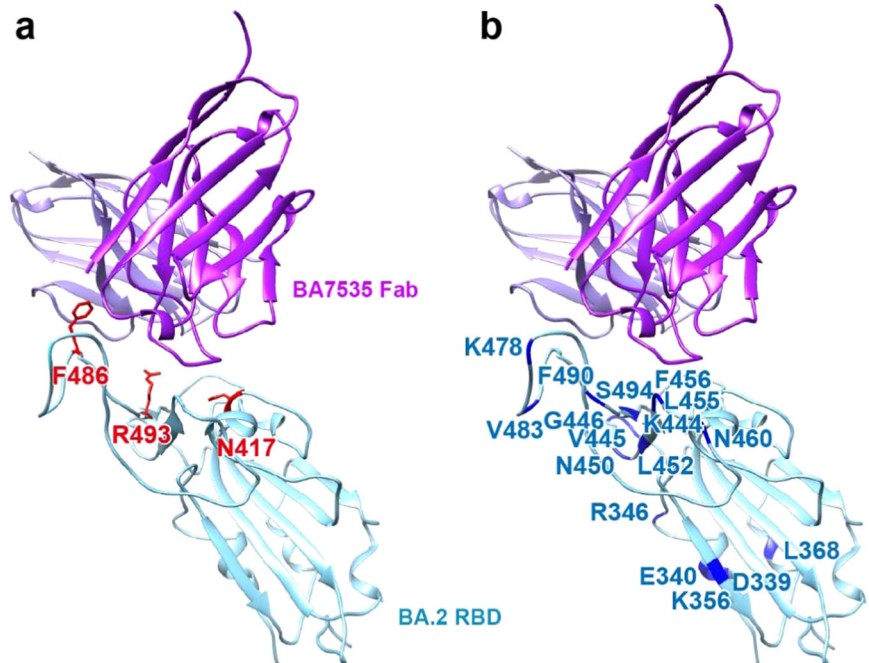

**Fig. 4 | BA7535 binding site avoids most mutations from Omicron sub-lineages.** The complexes between BA7535-Fab (purple) and RBD of Omicron sub-lineages (cyan). **a** Three mutation residues which are located close to the BA7535 binding site are highlighted in red. **b** Mutation residues that are not located to the BA7535 binding site are highlighted in blue.

## BA7535 targeting epitope is highly conserved and resists immune escape from recently emerged Omicron sub-lineages

Omicron harbors multiple mutations in the Spike protein, and these mutations are the key factors for the Omicron immune evasion. Next, we analyzed the mutations in the RBD from 42 sub-lineages of Omicron (Supplementary Table 5). Most of the mutations are not located in the BA7535 binding site except F486, for which four mutations, F486V, F486S, F486I, and F486P, are present in 24 Omicron sub-lineages. Other mutations, including N417T in BA.2.38.1 and R493Q in 30 Omicron sub-lineages, may also interrupt the BA7535 binding (Supplementary Table 5 and Fig. 4). We used PISA to analyze the hydrogen bond and the salt bridge possibly formed upon the above six mutations. As shown in Supplementary Table 6, we did not find a significant change in the binding interface area caused by the mutations, and the number of the hydrogen bonds and salt bridge are unchanged except the R493Q (Fig. 4a), which reduces one hydrogen bond and one salt bridge. Because the interface area avoids most of these mutation sites (Fig. 4b), antibody BA7535 should have a broad spectrum against most of Omicron variants. In addition, the $IC_{50}$ analyses (Fig. 1b) have indicated that the neutralizing activity of BA7535 against the sub-lineages containing the back mutation R494Q was not affected but the activity against the sub-lineages BQ.1, BQ.1.1, CH.1.1 and XBB was slightly impaired. Sequence analyses suggest that mutation N460K maybe the key factor (Fig. 1a). In addition, we only identify four hydrogen bonds within the three interfaces, the hydrogen bonds D420-Y106 and Y421-L103 have vanished. Most likely, N460K causes a local conformational change to D420 and Y421, resulting in the inability to interact with BA7535 to form hydrogen bonds, and ultimately, leads to a decline in the neutralizing activity of BA7535 against variant strains BQ.1, BQ.1.1, and XBB. In addition, based on the large interface area and remaining five hydrogen bonds, the BA7535 antibody retains its high binding affinity to all 42 sub-lineages, which may provide the molecular basis for efficient targeting of Omicron RBD by BA7535. To determine the capacity of BA7535 tolerant to existing and potential variants, all the BA7535-interacting residues on RBD protein were conducted for mutation assessment based on the SARS-CoV-2 uploaded in the GISAID dataset. Taking L452 site as a control, whose mutation frequency is nearly 56% and located outside the footprint of BA7535, the mutation frequency of all the antibody BA7535-interacting residues is lower than 0.5%, except Q493, which does not affect the neutralization activity (Supplementary Fig. 11). In summary, antibody BA7535 has a broad spectrum against most of Omicron variants.

### Pharmacokinetic and protective efficacy studies in mice

The pharmacokinetics of BA7535 was studied in three BALB/c mice with a single intravenous injection dose of 10 mg/kg. ELISA was used to monitor the antibody concentration in serum. Following a single dose, BA7535 showed satisfactory half-life and $AUC_{(0-t)}$, with a terminal half-life ($t_{1/2}$, $\lambda_z$) of ~95 h and $AUC_{(0-t)}$ of approximately 12785 h*µg/mL (Supplementary Fig. 12). Then we evaluated the prophylactic and therapeutic activity of BA7535 alone or in combination against challenge with Omicron BA.5 variant in K18-hACE2-transgenic mouse model[42]. BA7535 was administered at 2 or 10 mg/kg via intraperitoneal (i.p.) injection 24 h pre-infection or 8 h post infection. Lungs and brains were collected 2 and 4 days later for quantification of replicating virus, respectively. As shown in Fig. 5a–c, administration of BA7535 at 2 or 10 mg/kg reduced the lung viral titer by approximately 2.5 orders of magnitude, compared to a control. Complete abrogation of virus replication was observed in both lungs and brains. Similarly, BA7535 in combination with BA7208 (each mAb at 1 or 5 mg/kg) completely abrogated BA.5 virus replication in both lungs and brains, and animals receiving the mAb cocktail appeared to benefit from the additive contribution of the BA7208 mAb, as the mAb (BA7535/BA7208) cocktail treated animals maintained longer survival duration whereas those control mice all dies by round day 5 (Supplementary Fig. 13). It is worth noting that treatment with neutralizing

antibody failed to prevent weight loss and death in vivo challenged with high dose of SARS-CoV-2 Omicron BA.5 ($1 \times 10^5$ FFU) in agreement with previous reports[43,44], which suggest that neutralizing antibodies can be overwhelmed by a sufficiently high virus inoculum. Analysis of hematoxylin and eosin-stained lung sections from K18-hACE2 transgenic mice infected with SARS-CoV-2 displayed lung pathology with increased inflammatory cells around blood vessels and branches, and prominent inflammatory cells infiltration (Fig. 5c). Alleviated lung injury was observed in the high and low-dose BA7535 treated groups compared with the control group. In summary, BA7535 protected K18-hACE2 transgenic mice from SARS-CoV-2 BA.5 infection whether used alone or in combination with BA7208.

It has been reported that nanobodies or conventional antibodies can be delivered via intranasal and aerosol inhalation, which could be a more promising alternative way for delivering neutralizing antibodies[30,45]. To test this hypothesis, BALB/c mice were treated with BA7535 intraperitoneally at 10 mg/kg, intranasally at 1 mg/kg or through aerosol inhalation at 3 mg/kg 24 h before or after SARS-CoV-2 XBB.1 challenge (Fig. 5d). Lungs were harvested at day 2 post infection for quantification of viral titers by focus-forming assay (FFA). As shown in Fig. 5e–g, no live viral particle was detected in either the prophylactic or therapeutic group, whereas all animals in the control group revealed high viral titers in the lungs. BA7535 protected BALB/c mice from SARS-CoV-2 XBB.1 infection via diverse administration routes.

### Serial passage did not generate escape mutants in the presence of BA7535/BA7208 cocktail

The rational design of broad-spectrum neutralizing antibodies is to avoid or reduce the occurrence of antibody escape mutants. Thus, we next performed selection of escape mutants under pressure of diluted single antibodies and antibody combinations using SARS-CoV-2 virus (Fig. 6a). Herein authentic SARS-CoV-2 BA.5 was serially passaged with three-fold diluted BA7535, BA7208, or BA7535/BA7208 cocktail in Vero E6 cells, a cell line support SARS-CoV-2 replication. Because these neutralizing mAbs target the spike, we focused on mutations that occur in the spike. As shown in Fig. 6a–c in the presence of BA7535 the escape mutant N455K emerged after the third passage and became readily fixed in the population by the fourth passage, representing 100% of sequencing reads. For BA7208, the escape mutants D193V and L436R were dominant after the fourth passage, representing 86.9% and 88.4% of sequencing reads, respectively. Under the pressure of BA7535/BA7208 cocktail, the highest antibody concentration of cytopathic effect (CPE) was the same after three passages, and no obvious mutant viruses were observed by passage 4. The authentic virus neutralization assay suggested that N455K, D193V/L436R decreased the neutralizing ability of BA7535 and BA7208, respectively (Fig. 6d). In contrast, the BA7535/BA7208 cocktail group displayed similar qualitative percentages of CPE within both passage 1 and passage 3, and no dominant escape mutants were observed in the spike protein, indicating that the BA7535/BA7208 cocktail lead to less selective pressure compared to single-antibody treatment.

## Discussion

More new variants of SARS-CoV-2 arose under the pressure of humoral immunity established through vaccination and omicron infection and continued to impair the effectiveness of existing antibodies and vaccines. Both XBB and BQ.1.1 are completely resistant to LY-CoV1404, meaning there are no clinically authorized therapeutic antibodies effective against the emerging variants[46]. Seldom highly conserved neutralizing epitopes were identified on Omicron sub-lineages spike protein.

Here, we identified a highly potent neutralizing antibody BA7535 through extensive screening efforts using human antibody transgenic mice BA-huMab followed by phage display. BA7535 showed neutralization potency and breadth against not only previous variants,

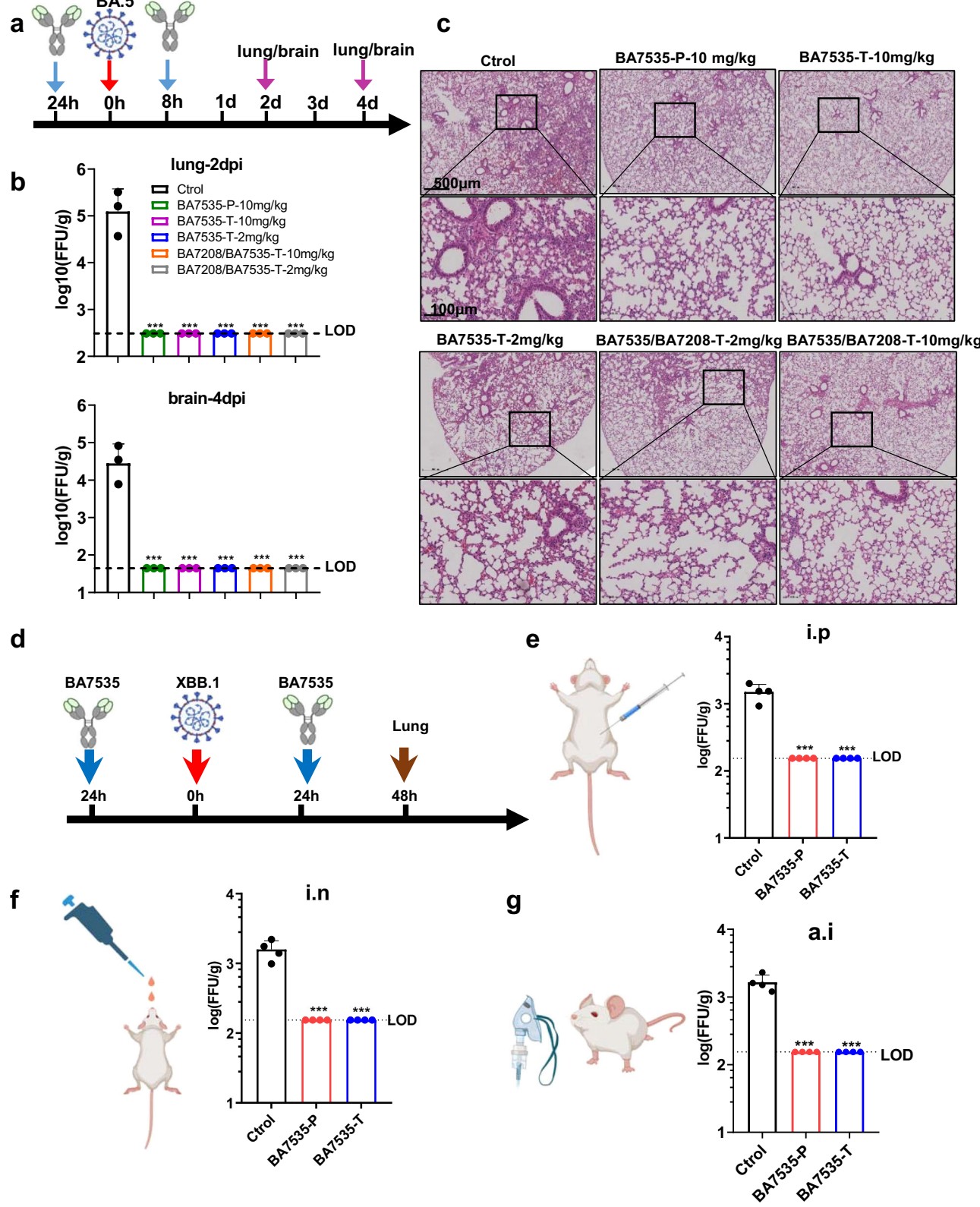

such as Alpha, Beta, and Omicron BA.1-BA.5 variants, but also all tested recently emerged Omicron subvariants, including BA.2.38.1, BF.7, BA.4.6, BA.5.5.1, XBB.1.5, CH.1.1, XBB.1.9.1 and EG.5, which suggests that there is still conserved antigenic epitope on RBD which remains unchanged during 3 years of evolution while maintains the ability to elicit neutralizing antibodies. BA7535, in combination with BA7208 provided higher resistance to immune escape by most SARS-Cov-2

variants. The neutralization potency of BA7535 alone or in combination with BA7208 was also verified in the mice model with Omicron BA.5 variant infections.

Despite the structural changes and significant antigenic shifts caused by the mutations in Omicron spike, the epitope for BA7535 is retained in most of Omicron and other VOCs during viral evolution, including the predominant variants of concern and emerging variants.

**Fig. 5 | BA7535 protected mice from Omicron BA.5 and XBB.1 variant infection.**
**a–c** Experimental route of prophylactic and therapeutic tests of BA7535 and
BA7535/BA7208 against BA.5 in hACE2 transgenic mice. **a** The timeline of neu-
tralizing antibody treatment and SARS-CoV-2 challenged is shown. Antibody was
administrated to hACE2 transgenic mice intraperitoneally (i.p) 24 h before (pro-
phylactic, abbreviation P) or 8 h after (therapeutic, abbreviation T) SARS-CoV-2
Omicron BA.5 infection ($1 \times 10^5$ FFU, intranasally). **b** Viral burden in the lungs and
brains was analyzed at 2 and 4 dpi by focus-forming assay (FFA) for infectious virus,
dashed line represents the limit of detection (LOD). The average ± SD of $n = 3$ from
two independent experiments is shown. Statistical significance was measured by
using one-way ANOVA with Dunnett's multiple comparisons, not significant [ns]:
$p > 0.05$, *$p < 0.05$, **$p < 0.01$, ***$p < 0.001$. **c** Pathological changes of H&E-stained
lung sections collected at 4 dpi from the prophylactic and therapeutic groups.
Alleviated lung injury was observed in the high- and low-dose prophylactic groups
compared with the PBS group. Images show low- (up; scale bars, 500 μm), and high-
power magnification (down; scale bars, 100 μm). Representative images from $n = 4$
per group. **d–g** Experimental route of prophylactic and therapeutic tests of BA7535
in BALB/c mice against XBB.1 in BALB/c mice. **d** The timeline of neutralizing

antibody treatment and SARS-CoV-2 challenged is shown. **e** BA7535 was admini-
strated to BALB/c (wt) mice (6–8 weeks old, female, $n = 4$ mice/group) intraper-
itoneally (i.p) at a dose of 10 mg/kg 24 h before (prophylactic, abbreviation P) or
after (therapeutic, abbreviation T) SARS-CoV-2 Omicron XBB.1 infection ($1 \times 10^5$
FFU, intranasally), the average ± SD from two independent experiments is shown.
**f** BA7535 was administered via intranasal (i.n) at a dose of 1 mg/kg 24 h before
(prophylactic, abbreviation P) or after (therapeutic, abbreviation T) SARS-CoV-2
Omicron XBB.1 infection ($1 \times 10^5$ FFU, intranasally, $n = 4$ mice/group), the aver-
age ± SD from two independent experiments is shown. **g** BA7535 was administered
via aerosol inhalation (a.i) at a dose of 3 mg/kg 24 h before (prophylactic, abbre-
viation P) or after (therapeutic, abbreviation T) SARS-CoV-2 Omicron XBB.1 infec-
tion ($1 \times 10^5$ FFU, intranasally, $n = 4$ mice/group). Viral burden in the lungs was
analyzed at 2 dpi by focus-forming assay (FFA) for infectious virus, dashed line
represents the limit of detection (LOD). PBS was administrated as a negative con-
trol, the average ± SD from two independent experiments is shown. **e–g** Statistical
significance was measured by using one-way ANOVA with Dunnett's multiple
comparisons, not significant [ns]: $p > 0.05$, *$p < 0.05$, **$p < 0.01$, ***$p < 0.001$. Source
data are provided as a Source Data file. Created with BioRender.com.

Structural analysis of the Omicron spike trimer with BA7535-Fab using
cryo-EM indicated that BA7535 avoided most of the mutational resi-
dues in RBD of diverse variants. Structural analysis is verified as an
efficient method to predict the immune escape ability of new variants
to anti-SARS-CoV-2 antibodies[28]. Based on the large interface area and
remaining five hydrogen bonds, the BA7535 antibody still retains the
high binding affinity to all 42 Omicron sub-lineages, which may provide
the molecular basis for efficient targeting of Omicron RBD by BA7535.
The BA7535 shared a partial epitope with Class I antibodies. These
overlapping epitope residues are far from the mutation sites in the
Omicron RBD.

Structure analysis indicated that the epitope targeted by BA7535
was highly conserved among SARS-CoV-2 variants. It is difficult to
predict whether the conserved epitope-targeted antibody or vaccine
will be escaped in the future. The mutation probability of conserved
epitope was determined by many factors, including virus structural
stability, infectivity, spike-ACE2 binding affinity, host response, and
selective pressure. Some conserved residues are essential to ACE2
binding or structural stability. SASR-CoV-2 uses RBD to interact with
host angiotensin-converting enzyme 2 (ACE2) and ensure cell recog-
nition. The epidemic mutations always influence the virus infectivity or
fitness. Meanwhile, most variants convergently acquired similar amino
acid substitutions at critical residues in the spike. Epidemic dynamics
modeling also suggests that most hot mutations or substitutions
increase viral fitness. All of the above indicated that it is difficult to
mutate the conserved epitope involving viral structure or stability.

BA7535 exhibited exceptional breadth and was able to neutralize
all tested variants No one variant tested was observed to escape from
neutralization. Though SARS-CoV-2 reference strain and D614G strains
only impaired the potency of BA7535, the combination of BA7535 and
BA7208 overcame immune escape and provided broader neutralization
potency against SARS-CoV-2 wt and D614G variant than alone. As
for the generation of SARS-CoV-2 escape mutations against BA7535
during serial passage, the escape mutant N455K emerged in the pre-
sence of BA7535 after the third passage and became readily fixed in the
population by the fourth passage. However the $IC_{50}$ of BA7535 only
increased from 2.8 ng/mL to 14.6 ng/mL and still performed potent
neutralizing activity, the mutation N455K did not contribute to the
immune escape sharply. We also detected the the binding affinity of
BA7535 to BA.5 RBD and mutant RBD (N455K); to some extent, there
was mild decrease in binding affinity of mutant RBD (N455K) (KD =
1.625E-8) against ACE2 compared with reference strain RBD(KD =
5.586E-9), which may influence the fitness (Supplementary Fig. 14).
Apart from that, serial passage did not generate escape mutants in the
presence of BA7535/BA7208 cocktail, which clear the way for clinical
application of BA7535.

Neutralizing monoclonal antibodies (mAbs) targeting the RBD can
inhibit viral infection by blocking receptor engagement. Nevertheless,
some antibodies that bind to SARS-CoV-2 epitopes can also mediate
protection from disease by activating immune effector cells through
interactions between the immunoglobulin Fc region and Fc receptors
(FcRs), resulting in the clearance of infected cells, including Ab-
dependent cellular cytotoxicity (ADCC)[47] and Ab-dependent cellular
phagocytosis (ADCP)[48]. BA7208 displayed superior ability to mediate
ADCC and ADCP, while BA7535 and LY-CoV1404 showed no or little
ability to mediate ADCC or ADCP. Previous reports indicated that Fab
and Fc functions may be considered interdependent. The epitope[49],
specificity[50], affinity, and antibody concentration[51] all affect antibody
functional response activation. In the in vivo protection assay, the
combination of BA7535 and BA7208 low-dose group performed better
than alone in alleviating weight loss. The ADCC and ADCP may play
additional role in the vivo protection, which need further confirmation.

As for the antibody gene usage, we also analyzed the genetic
feature of mAb BA7535 and revealed that BA7535 was derived from the
pairing of IGHV3-30 and IGKV5-2. Germline usage analysis of 5712 SARS-
CoV-2 spike-specific human mAbs from the COV-AbDab database[52]
shows that the BA7535 class of antibodies with IGHV3-30/IGKV5-2
pairing is rare in the human antibodies (Supplementary Fig. 15), sug-
gesting that this class of antibodies does not appear to be induced
preferentially in human population by SARS-CoV-2 infection or vacci-
nation. This is one of the advantages of using humanized mouse
immunization and phage display technology, which allows us to screen
for broad neutralizing antibodies that are rare in humans.

In summary, we identified a potent broadly neutralizing antibody
BA7535 targeting a highly conserved epitope on the RBD from pre-
vious and recently emerged SARS-CoV-2 variants. Human antibody
BA7535 effectively neutralizes all tested previous and current emer-
ging SARS-CoV-2 variants ranging from 2019 to 2023. BA7535 can
protect hACE2-transgenic mice from Omicron BA.5 infections, thus
making it a promising candidate as an intervention against constantly
emerging SARS-CoV-2 variants.

## Methods
### Cell lines
Cell lines used in this study were obtained from ATCC (HEK293T, Vero
E6, Vero CCL81 cells) or Invitrogen (Expi-293F cells). All cell lines used
in this study were routinely tested for mycoplasma and found to be
mycoplasma-free.

### Ethical statement
All animal experiments complied with relevant ethical regulations
regarding animal research. Immunization and pharmacokinetics study

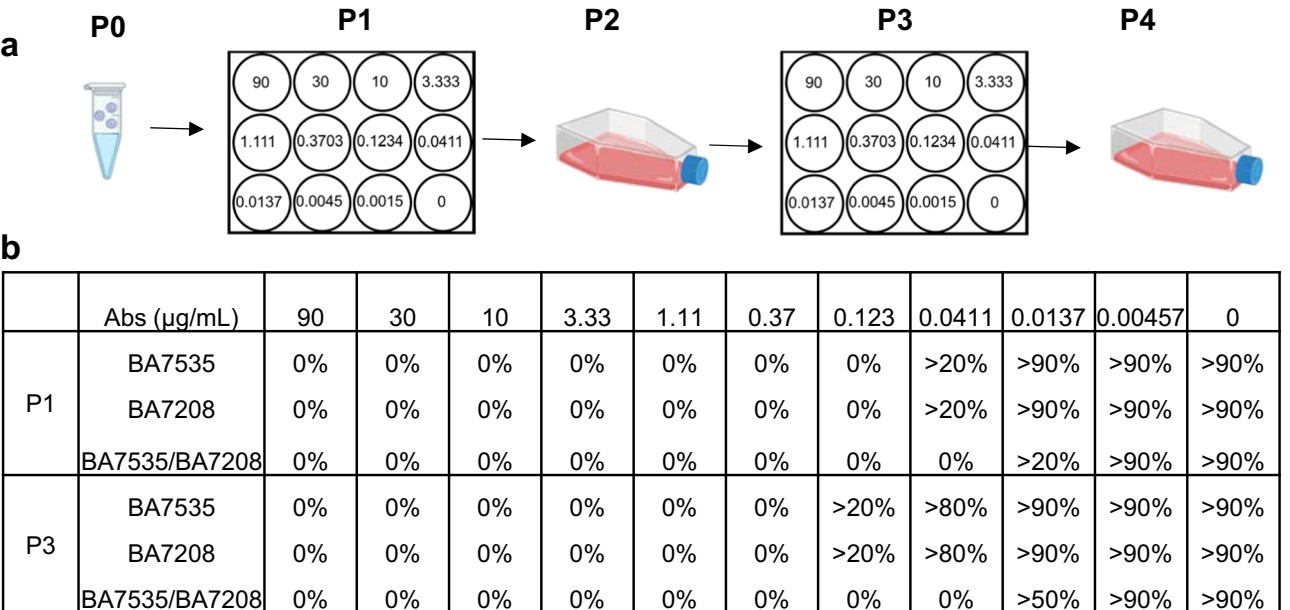

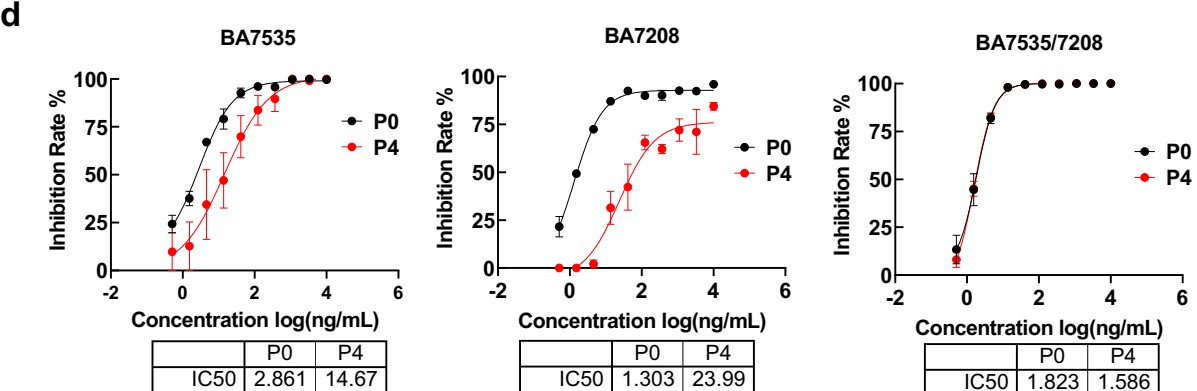

**Fig. 6 | Escape mutant screening under the pressure of antibodies. a** Schematic of the escape mutant screening process. Authentic SARS-CoV-2 BA.5 was passaged in the presence of antibodies (BA7535, BA7208, BA7535/BA7208 cocktail) with serial dilutions on Vero E6 cells. CPE was used to monitor virus replication. **b** Quantitative percentage of CPE under different concentrations of antibodies during passage 1 (P1) and passage 3 (P3). **c** Next-generation sequencing of virus RNA from passage 1, 3, and 4. Variant nucleotides and corresponding residues are shown, and the qualitative percentage of variant residues observed in each dilution is presented. **d** Neutralization of antibodies (BA7535, BA7208 and BA7535/7208) against authentic virus passage 0 (P0) and passage 4 (P4), the average ± SD from two independent experiments is shown. The experiment was performed with duplicate samples. Data are average values of two replicates. Source data are provided as a Source Data file. Created with BioRender.com.

procedures in mice were approved by the Institutional Animal Care Committee of Boan Biotech, and the Approval Numbers are 2021-TS0001-36 and 2022-TS0001-21, respectively. The animal study was reviewed and approved by the Institutional Animal Care and Use Committees of the First Affiliated Hospital of Guangzhou Medical University (2021-239).

## Immunization

Six human antibody transgenic mice BA-huMab (ID Q44, Q45, Q46, Q47, Q48, and Q49) were immunized with 35 μg recombinant RBD protein of SARS-CoV-2 BA.1 for three rounds and one final boost in 10-day intervals. Freund complete adjuvant was used in 1st round, Freund incomplete adjuvant was used in 2nd–3th rounds, and no adjuvant was used in the final booster. Spleen cells were harvested after three days of the last boost for phage library construction.

## Phage display library construction and screening

RNA was extracted from spleen cells of each immunized mouse in a Trizol method separately. After cDNA synthesis, the variable regions of the heavy and light chains were amplified separately and integrated into pCOMB3x vector, and then the products were electro-transfected into *Escherichia coli* TG1 for preparation of phage library. Spike RBDs of BA.1 were used for the panning of libraries. Plates coated with 5 μg/mL protein or 20 μL streptavidin-magnetic beads loading 5 μg biotinylated protein were used to capture phages with interest ScFvs. After incubation with input phages at 37 °C for 2 h and washing 5–6 times with PBST (PBS containing 0.05%Tween-20), captured phages were eluted and used to infect *E. coli* TG1. After 3 rounds panning, TG1 cells infected eluted phages were grown on plates. ScFvs were expressed, and positive hits were obtained and sequenced. RNA was extracted from the spleen cells of each immunized mouse by Trizol method separately. cDNA synthesis was performed using Transcriptor First Strand cDNA Synthesis Kit. The variable regions of the heavy and light chains were amplified from the cDNA by PCR separately and were integrated into the pCOMB3x vector, and then the products were electro-transfected into *Escherichia coli* TG1 for preparation of phage library.

## Human monoclonal antibodies

Heavy-chain variable region and light-chain variable region of the recombinant antibody were amplified (2× Phanta Max Master Mix, Vazyme, P515-01) using the positive clones as the template. Heavy-chain variable region was fused (ClonExpress II One Step Cloning Kit, Vazyme, C112-01) into the linearized pcDNA3.4 vectors with human IgG1 constant region. Light chain variable region was fused (ClonExpress II One Step Cloning Kit, Vazyme, C112-01) into the linearized pcDNA3.4 vectors with human κ constant region. Antibodies were expressed with Expi-CHO Expression system (Gibco) for 10–12 days, and the supernatants were harvested and purified by protein A resin (BestChrom, AA0272) and Chromdex 200 PG resin (BestChrom, AG0083) sequentially.

## ELISA-based receptor-binding inhibition assay

High-binding ELISA plates were coated with 0.125 μg/mL recombinant SARS-CoV-2 RBD of SARS-CoV-2 BA.1 at 4 °C overnight, and then blocked with 3% skim milk powder in PBST at 37 °C for 1 h. ScFv or serial diluted antibody was mixed with biotinylated human ACE2 (final concentration 0.04 μg/mL), and then the mixture was incubated with coated RBD in the plates for 1 h at 37 °C. After washing two times, the retained biotinylated ACE2 binding to coated RBD was detected by HRP-conjugated Streptomycin.

$$\text{nhibition rate\%} = (OD_{450} \text{ of no antibody} - OD_{450})/OD_{450} \text{ of no antibody} \times 100\%$$

$$(1)$$

## Bio-Layer Interferometry (BLI) based competitive binding assay

Competitive binding of the antibodies was performed on a ForteBio Octet Red96 system (Pall Forte BioCorporation, Menlo Park, CA) using in-tandem format binning assay. Biotinylated RBD protein of Omicron BA.1 variant (Sino Biological, Cat. 40592-V08H121) was loaded onto SA sensors (Fortebio, Cat. 18-5019). The sensors were then exposed to the first antibody with 30 μg/mL or PBST for 100 s, then to the second antibody at 30 μg/mL for 100 s. Data was processed using ForteBio's Data Analysis Software 9.0.

$$\text{Inhibitory rate (IR)} = \left(1 - \frac{\text{Response of Antibody}}{\text{Response of Mock}}\right) \times 100\% \quad (2)$$

## Pseudovirus neutralization assay

Vesicular Stomatitis Virus (VSV) pseudotyped with SARS-CoV S protein provided by Beijing SanYao Science & Technology Development Co. were produced and titrated as below. 50 μL SARS-CoV-2 pseudovirus ($1.3 \times 10^4$ TCID$_{50}$/mL) were incubated with threefold serially diluted antibody at 37 °C for 1 h, and then 100 μL cell suspension of Huh-7 were added to the mixtures. After 24 h incubation at 37 °C, 150 μL supernatant was abandoned, and 100 μL Bright-Glo was added as a substrate. Luminous value was detected by Microplate reader (EnVision, PE), and inhibitory rate was calculated as below.

$$\text{Inhibitory rate\%} = (1 - (\text{mean RLU of sample} - \text{mean RLU of blank control})/$$
$$(\text{mean RLU of negative control} - \text{mean RLU of blank control}) \times 100\%)$$

$$(3)$$

The half-maximal inhibitory concentrations (IC$_{50}$) were determined using 4-parameter logistic regression (GraphPad Prism). Experiments were performed in duplicate, value = Mean ± SD.

## Focus reduction neutralization test (FRNT) of SARS-CoV-2

FRNT assay was used for the evaluation of the monoclonal antibody neutralization effect as previously reported with some modification[53]. Vero E6 cells were seeded into a 96-well plate one day before infection. The next day, serially diluted monoclonal antibody BA7535, BA7208 or cocktail (BA7535/BA7208) and SARS-CoV-2 BA.1, BA.2, BA.5, XBB.1, XBB.1.5, XBB.1.9.1 and EG.5 (120–150 FFU, Focus-forming unit) were combined in DMEM containing 2% FBS and incubated at 37 °C for 1 h. Then, 50 μL mixtures were added into 96-well plate seeded with Vero E6 cells and incubated at 37 °C for 1 h. Inoculums were removed before adding the overlay media (100 μL MEM containing 1.2% carboxymethylcellulose). The plates were then incubated at 37 °C for 24 h. Overlays were removed and cells were fixed with 4% paraformaldehyde solution for 30 min. Cells were permeabilized with 0.2% Triton X-100 and incubated with cross-reactive rabbit anti−SARS-CoV-N IgG (Sino Biological Inc., catalog 40143-R001) for 1 h at room temperature before adding HRP-conjugated goat anti−rabbitIgG (H + L) antibody (1:4000 dilution) (Jackson ImmunoResearch, catalog 111-035-144). Cells were further incubated at room temperature. The reactions were developed with KPL TrueBlue Peroxidase substrates (Seracare Life Sciences Inc.). The number of SARS-CoV-2 foci was calculated using an EliSpot reader (Cellular Technology Ltd.). The IC$_{50}$ is determined by 50% focus reduction neutralization test titers (FRNT$_{50}$) which was used for evaluation of the potency of hACE2-Fc in inhibiting SARS-CoV-2 replication.

## Affinity to SARS-CoV-2 Spike RBD from SARS-CoV-2 variants

The binding kinetics were assessed by Surface Plasmon Resonance (SPR) assay using the BIAcore 8 K system. HBS-EP$^+$ buffer (150 mM NaCl, 10 mM HEPES, 3 mM EDTA, and 0.05% (v/v) surfactant P20 pH

7.4) was used as a running buffer. The blank channel of the chip served as the negative control. Antibody was captured on ProA chip at 400–500 response units. Serial dilutions of SARS-CoV-2 BA.1 RBD proteins (from 50 nM to 3.125 nM with 2-fold dilution) in running buffer were applied to flow over the chip surface. After dissociation for 450 s, the chip was regenerated for 30 s with 10 mM pH 1.5 Glycine after each cycle. The affinity was calculated using a 1:1 (Langmuir) binding fit model with BIAevaluation software. Experiments were performed in duplicate, value = Mean ± SD.

### ADCC
ADCC reporter bioassay was conducted with CHO-K1-Spike cells as target cells and Jurkat cells (G7011, Promega) as effector cells. Target cells, effector cells, and serially diluted antibodies were mixed in white 96-well plates and incubated in the cell incubator (37 °C, 5% CO2) for 6 h. Then Bio-Lite Chromogen Solution was added and incubated at Room Temperature (RT) for 15 min. Plates were read by Tecan Microplate Reader. Experiments were performed in duplicate, value = Mean ± SD.

### ADCP
For ADCP, CHO-K1-Spike cells were used as target cells, and Jurkat-FcγRIIA-H131 cells (Vazyme) were effector cells. Target cells, effector cells, and serially diluted antibodies were added and mixed in white 96-well plates and incubated in the cell incubator (37 °C, 5% $CO_2$) for 6 h. Then Bio-Lite Chromogen Solution was added and incubated at RT for 2–5 min. Plates were read by Tecan Microplate Reader. Experiments were performed in duplicate, value = Mean ± SD.

### Cell-based binding
CHO-K1 cells expressing SARS-CoV-2 Spike were cultured for two passages and then harvested, followed by washing twice with FACS buffer (0.2% BSA in PBS). The CHO-K1/Spike cells were stained with gradient diluted antibody at 4 °C for 1 h. After washing twice with FACS buffer, cells were incubated in the dark with 100 μL FITC-anti-human IgG Fc in 1 μg/mL at 4 °C for 30 min. Cells were washed twice and then resuspended in 100 μL FACS buffer for analysis by NovoCyte 2060 R flow cytometry. Experiments were performed in duplicate, value = Mean ± SD.

### Cryo-EM sample preparation and data acquisition for the Omicron BA.2 S-BA7535 complex
20 μL purified Omicron BA.2 Spike protein at the concentration of 2.3 mg/ml was incubated with 5 μL BA7535-Fab at the concentration of 3.2 mg/mL at a 1:3 molar ratio on ice for 20 min. The complex was applied for Cryo-EM grid preparation. Quantifoil Au R2/1 grids were first to glow discharged for 60 s using a Pelco easiGlow glow discharged unit. An aliquot of 3.5 μL protein sample of Omicron BA.2 S-BA7535 complex was applied to the surface of the grid at a temperature of 6 °C and a humidity level of 100%, blotted with filter paper for 3.5 s before being plunge-frozen in liquid ethane using Vitrobot Mark IV (Thermo Fisher Scientific). Cryo-EM micrographs were collected on a 300 kV Thermo Fisher Scientific Titan Krios G3i electron microscope equipped with a GIF-Quantum energy filter (Gatan), which was used with a slit width of 20 eV. Automatic data collection was performed using EPU software. Images were recorded with Gatan K3 direct electron detector. The micrographs were collected at a calibrated magnification of ×130,000, yielding a pixel size of 0.668 Å at a super-resolution mode. In total, 8789 micrographs were collected at a dose rate of 15e-/pixel/s and an accumulated electron dose of ~50 e−/Å$^2$ on each micrograph that was fractionated into 36 movie-frames. The final defocus range of the datasets was approximately −(1.2–2.0) μm for the Omicron BA.2 S-BA7535 datasets (see detailed data collection parameters in Supplementary Table 1).

### Image processing and 3D reconstruction
Beam-induced motion correction was performed on a stack of frames using MotionCor2. Initial contrast transfer function (CTF) values for each micrograph were calculated with CTFFIND4. Micrographs with an estimated resolution limit worse than 4 Å were discarded in the initial screening. A total of 8789 good micrographs were selected for further data processing and reconstruction using cryoSPARC. 4,038,887 particles were picked from the selected micrographs. Then, the picked particles were extracted and subjected to three rounds of reference-free 2D classification in cryoSPARC. 1,564,880 particles were selected from good 2D classes and were subjected to one round of hetero refinement. The subsets were subjected to one round of 3D classification. 920,863 particles from four classes showing high resolution were selected for reconstruction. Next, a density map with global resolution of 2.40 Å was obtained after 3D refinement using particles selected from 3D classification. The density quality of S-BA7535 complex was good enough for model building except for the RBDs and BA7535-Fabs. After one round of local refinement focused on the RBD/ BA7535-Fab with mask, the densities of RBD/ BA7535-Fabs were refined to 3.07 Å. Local resolution estimate was performed with cryoSPARC (see detailed workflow in Supplementary Fig. 1).

### Model building
The model of Omicron BA.2 S-BA7535 complex was built based on the model of Delta Spike protein in complex with BA7208-Fab and BA7125-Fab (PDB:7XDL). The model was manually modified in COOT. The modified model was refined using phenix.realspace_refine program in PHENIX software package. Statistics for model refinement and validation are listed in Supplementary Table 1.

### Pharmacokinetics studies
A single intravenous injection of BA7535 was conducted in BALB/c mice ($N = 3$, 3/group, female, age 7-8 weeks, body weight 20 ± 1 g) at 10 mg/kg. Blood samples were collected at predose and 5 min, 1 h, 6 h, 24 h, 72 h, 120 h, 168 h, 240 h, 336 h postdose from the mice via orbit vein bleeding. ELISA was used to determine the concentration of BA7535 in serum. RBD protein of SARS-CoV-2 BA.1 variant was used as the capture reagent, and goat anti-human IgG was the detecting agent. The main PK kinetic parameters were calculated using Phoenix WinNonlin. Values are shown as Mean ± SD.

### Protection against Omicron BA.5 in K18-hACE2-transgenic mice via the intraperitoneal route
The in vivo prophylactic and therapeutic potency of BA7535 and BA7535/7208 cocktail against SARS-CoV-2 Omicron BA.5 were evaluated in K18-hACE2-transgenic female mice, 6- to 8-weeks old mice were intraperitoneally injected with 2 or 10 mg/kg of BA7535 or BA7535/ BA7208 cocktail per mouse 24 h before or 8 h after the challenge with $1 \times 10^5$ FFU SARS-CoV-2 Omicron BA.5. Mice injected with phosphate-buffered saline (PBS) were challenged with the same dose of SARS-CoV-2 as control. To investigate the presence of SARS-CoV-2 in the lungs and brains, lungs and brains were harvested for viral titers 2 and 4 days later by using a focus-forming assay (FFA). Meanwhile lung tissues were collected and stained for histopathological analysis, weight change was monitored. Weight loss and survival rate were monitored every day. The SARS-CoV-2 Omicron BA.5 strain was presented by the Guangdong Provincial Center for Disease Control and Prevention, China. Experiments related to authentic SARS-CoV-2 were conducted in Guangzhou Customs District Technology Center ABSL-3 Laboratory.

### Protection against Omicron XBB.1 in BALB/c mice via diverse administration routes
The prophylactic and therapeutic efficacies of BA7535 against SARS-CoV-2 Omicron XBB.1 were evaluated in BALB/c female mice, six- to

8-weeks old mice were intraperitoneally injected with 10 mg/kg of BA7535 per mouse 24 h before or after the challenge with $1 \times 10^5$ FFU SARS-CoV-2 Omicron XBB.1. Mice injected with PBS were challenged with the same dose of SARS-CoV-2 as control. Lungs were harvested for viral titers 2 days later by using a focus-forming assay (FFA). Meanwhile, we used an aerosol inhalation apparatus to administer the antibody for evaluation of its protection in vivo. After administration of antibody via aerosol inhalation (3 mg/kg) or intranasal (1 mg/kg), we collected the lung 2 days after infection for viral titers. The SARS-CoV-2 Omicron XBB.1 strain was presented by Guangdong Provincial Center for Disease Control and Prevention, China. Experiments related to authentic SARS-CoV-2 were conducted in Guangzhou Customs District Technology Center ABSL-3 Laboratory.

### Escape mutant screening and high-throughput sequencing

The experiments were performed as previously reported[54] with some modifications. Briefly, SARS-CoV-2 BA.5 (P0) was incubated with a series of three-fold dilutions (ranging from 90 μg/mL to 0.00457 μg/mL) of each antibody for 1 h and then added to a monolayer of Vero E6 cells at a multiplicity of infection (MOI) of 0.01 focus-forming unit (FFU)/cell. After 1-hour incubation the supernatant was discarded and cells were washed with PBS three times, the media containing the diluted Abs were added to the wells. The cytopathic effect was monitored over 4 days, and supernatants (passage 1, P1) were collected from wells with the highest concentration of antibody with detectable CPE (≥20% CPE). Then 200 μl of the P1 supernatant was expanded in a T25 flask containing Vero E6 cells under the same diluted antibody pressure (P2). For the third-round passage, SARS-CoV-2 BA.5 P2 was incubated with a series of three-fold dilutions (ranging from 90 μg/mL to 0.00457 μg/mL) of each antibody for 1 h and then added to a monolayer of Vero E6 cells at a multiplicity of infection (MOI) of 0.01. The cytopathic effect was monitored over 4 days, and supernatants (P3) were collected from wells with the highest concentration of antibody with detectable CPE (≥20% CPE), 200 μL of the P3 supernatant was expanded in a T25 flask containing Vero E6 cells under the same diluted antibody pressure (P4). Viral RNA of P0-P4 were isolated with Viral RNA extraction kit (QINGEN) and used for high-throughput sequencing. CLC Genomics Workbench Version 21 was used for SNPs analysis. Single-nucleotide polymorphisms with a minimum frequency of 5% were identified and annotated.

### Statistics and reproducibility

All statistical analyses were performed in GraphPad Prism 8 software, except for affinity data, which was performed in Excel. Details on the statistical tests applied are provided in figure legends. The data are reported as bar graphs displaying individual values and means ± SD, as indicated in the figure legends. No experiments were excluded from the analyses.

### Reporting summary

Further information on research design is available in the Nature Portfolio Reporting Summary linked to this article.

## Data availability

All relevant data are available in the article, supplementary Information, or from the corresponding author J.Z., upon reasonable request. Cryo-EM density maps of the spike-Fab complex have been deposited in the Electron Microscopy Data Bank (EMD-34522 and EMD-34526). The atomic coordinates of the above complexes have been deposited in the Protein Data Bank under the accession codes 8H7L and 8H7Z, respectively. The sequences of BA7535 have been deposited in GenBank with the accession codes OP831943 and OP831944 for heavy chain and light chain, respectively. The other source data generated in this study are provided in the Supplementary Information and the Source Data file. Source data are provided in this paper.

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

## Acknowledgements

This project was supported by a grant from National Key R&D Program of China (2021YFC2300101 Y.W., 2023YFC3041600 Y.W., 2022YFC 0869200 C.D.), the National Natural Science Foundation of China (82172240 Y.W., 92369113 Y.W., 81870246 and 82070329 Z.L., 82025001 J.Z., 32000658 L.Z., 81971500 J.Z.), Self-supporting Program of Guangzhou Laboratory (SRPG22-001), Guangdong Basic and Applied Research Projects (2023B1515020040 Y.W., 2021B1212030016 J.D., 2021A1111100009 J.D.), ZhongNanShan Medical Foundation of Guangdong Province (ZNSA-2020013), State Key Laboratory of Respiratory Disease (SKLRD-Z-202411 L.Z., SKLRD-Z-202214 Y.W., SKLRD-OP-202309 Y.W.), Science and Technology Planning Project of Guangzhou City (2023A04J1279 L.Z.), Guangzhou Medical University (YP2022005 Y.W.), the Science and Technology Project of General Administration of Customs, P.R. China (2023HK065 L.Z.)and Korea Institute of Planning and Evaluation for Technology in Food, Agriculture and Forestry (IPET) through Animal Disease Management Technology Advancement Support Program, funded by the Ministry of Agriculture, Food and Rural Affairs (MAFRA) (122012-2, 122060-2) (J.Z.). We thank the Biobank for Respiratory Disease in the National Clinical Research Center for Respiratory Disease (BRD-NCRCRD, Guangzhou, Southern China).

## Author contributions

J.C.Z., Z.L., C.L.D., and L.W. initiated and coordinated the project. Y.W. and Lu.Z. evaluated the neutralizing potency in vitro and vivo. A.Y., Z.J., Y.G, R.Q., and X.M. performed cryo-EM studies. D.S. and Y.W. designed the experiments and wrote the manuscript. C.L.D. and Z.L. revised the manuscript. C.C.D. and J.F. did the antibody screening-based phage display. M.R. performed the affinity experiments. M.D. evaluated the neutralizing potency using pseudovirus. H.L. and R.P. constructed the antibodies. J.C., Z.Z., J.X., J.X.Z., and J.D. performed escape mutant screening under pressure of single antibodies and antibody combinations. B.Y. and Q.W. conducted antigen immunization. M.Z. and Li.Z. did the ELISA experiments. J.J. evaluated the ADCC and ADCP activity. P.X. performed the antibody purification. D.L. and B.S. performed the PK studies. J.S., Y.L., and Y.Z. performed cell culture for antibody expression. Z.S. conducted the antibody quality analysis and quality control.

## Competing interests

Patents have been filed for BA7535. All authors affiliated with Shandong Boan Biotechnology Co., Ltd are employees of Shandong Boan Biotechnology Co., Ltd (Yantai). The remaining authors declare no competing interests. Correspondence and requests for materials should be addressed to Jincun Zhao.

## Additional information

[1]State Key Laboratory of Respiratory Disease, National Clinical Research Center for Respiratory Disease, Guangzhou Institute of Respiratory Health, the First Affiliated Hospital of Guangzhou Medical University, Guangzhou, China. [2]Clinical Laboratory Medicine Department, The Second Affiliated Hospital of Guangzhou Medical University, Guangzhou, China. [3]GMU-GIBH Joint School of Life Sciences, Guangzhou Medical University, Guangzhou, China. [4]Cryo-electron Microscopy Center, Southern University of Science and Technology, Shenzhen, China. [5]Antibody Research and Development Center, Shandong Boan Biotechnology Co., Ltd., Yantai, China. [6]Division of Monoclonal Antibodies, Institute for Biological Product Control, National Institutes for Food and Drug Control (NIFDC), Beijing, China. [7]Health and Quarantine Laboratory, Guangzhou Customs District Technology Centre, Guangzhou, China. [8]Guangzhou National Laboratory, Bio-Island, Guangzhou, China. [9]Shanghai Institute for Advanced Immunochemical Studies, School of Life Science and Technology, ShanghaiTech University, Shanghai, China. [10]Institute for Hepatology, National Clinical Research Center for Infectious Disease, Shenzhen Third People's Hospital; The Second Affiliated Hospital, School of Medicine, Southern University of Science and Technology, Shenzhen, China. [11]These authors contributed equally: Yanqun Wang, An Yan, Deyong Song, Maoqin Duan. ✉e-mail: wanglan@nifdc.org.cn; douchanglin@boan-bio.com; liuz3@sustech.edu.cn; zhaojincun@gird.cn

