## [Peer Review File · Nature Communications]

Identification of a highly conserved neutralizing epitope within the RBD region of diverse SARS-CoV-2 variantsEditorial Note: Parts of this Peer Review File have been redacted as indicated to remove third-party material where no permission to publish could be obtained.

REVIEWER COMMENTS

Reviewer #1 (Remarks to the Author):

In comparing the binding interface to the other epitope classes, the authors superpose their complex with individual antibody complex structures, but neglect to discuss epitope classes commonly used in the literature:

Classes 1-4 described by Barnes et al Nature 2020 Dec;588(7839):682-687

Class 5 described by Starr et al Nature volume 597, pages 97–102 (2021)

Class 6 described by Rouet et al Nature Communications volume 14, 687 (2023)

The authors should clarify which of these previously described classes their BA7535 (and BA7208) share epitope surface with (given that the cryoEM structure shows overlap with the ACE2 surface, the epitope class is likely to be predominantly 1 or 2). Class 1 generally implies that the RBD must be in an up-conformation to allow binding, but have the authors investigated whether the binding surface might be available with the RBD in the down conformation?

Technical Comments

The authors need to release coordinates in the PDB so that these structures can be properly interrogated by reviewers.

The figures of the structures are poor: detail in Figure 3b, in particular, is almost uninterpretable (too many light shade pastels; poor resolution preventing meaningful enlargement).

Likewise, Figure 4 is similarly difficult to interpret due to low resolution. Figure 4 would be improved by only showing a handful of high-resolution images that explicitly highlight the point mutants in close proximity to the antibody interface.

The structure they present is one derived from cryo-EM, although the nominal resolution at the RBD-Fab interface is poor (4 Å), optimised to only about 3 Å. It worries me that they then plug this into PISA from which they derive all sorts of metrics – buried surface, deltaG and hydrogen bonds etc – which is ordinarily fine, but it is a red flag to me that they list a bunch of hydrogen bonds with distances that to my mind are too large to place any confidence in, especially given the low resolution of the model. To my mind, if the distance is larger than 3.2 Å, it's too far to be confidently ascribed as an H-bond, yet distances of 3.4, 3.5, 3.6 and 3.6 are listed. Additionally, in Table S3 they list buried surface to 4 significant figures (two or three would be more realistic given that the number can vary +/- 50 Å² by changing rotamer of a single residue). This is symptomatic of what I would call general over interpretation of low-resolution structural data and extends to the generation of in silico point mutants: it wasn't clear how these were derived and what rotamers were chosen.

Reviewer #2 (Remarks to the Author):

The manuscript by Wang and colleagues details a human monoclonal antibody (BA7535) with broad neutralisation activity against SARS-CoV-2 variants. Using a combination of vaccination of humanised mice, followed by phage display, the authors recovered BA7535 and confirmed broad neutralisation activity in vitro and in vivo. Potency was strongest against BA1-BA5 lineages, with significant loss of activity against ancestral and more recent strains. Structural analysis reveals an epitope at the top of the RBD and bioavailable in the "up" conformation. The paper is clearly presented, although the information gain from a single antibody epitope at this stage of the pandemic is somewhat limited.
Comments:

1 – This antibody was elicited by immunisation of a humanised mouse then phage display, which may limit applicability to human responses to SARS-CoV-2. Can the authors identify any human antibodies that bind with similar parameters (or share genetic features) in the extensive BCR sequencing or mAb datasets available? i.e. how common are such antibodies?

2- The ADCC and ADCP results are confounding given most spike-specific mAbs should give some signal in this assay. The authors should provide a rationale in the Discussion for why BA7535 and LY-CoV1404 fail to engage FcR.

3 – Line 230 - " the risk of being affected by mutation of RBD sites and enhance the broad spectrum while improving the neutralization efficiency. In vitro neutralizing potency experiments also confirmed the broad spectrum of BA7535 (Fig. 1a). In conclusion, among these antibodies mentioned above, BA7535 is an efficient one with broad neutralizationpotency"

What is meant by efficiency in this case? BA7535 neutralises with comparable or lesser potency than many of the comparator mAbs named.

4- Line270 - "In summary, antibody BA7535 has a broad spectrum against all Omicron variants."

This is overly strong and not confirmed by the data but predicted by the authors based on surface areas not experimental validation. The authors too concede that local conformational changes can affect hydrogen bonding between the epitope and paratope, as such this language should be more circumspect.

5 – Somewhat confoundingly, BA7535 treatment (either 10mg/kg or 2mg/kg) showed no protection against weight loss despite eliminating viral replication and spread. How do the authors account for this? Animals terminated by day 4, so it is difficult to ascertain a complete picture of pathogenesis. This study should be repeated with completed weight analysis until recovery or humane endpoint.

6- It is interesting the viruses so rapidly escaped (by P3-4) under antibody selection pressure. Are there any fitness cost associated with these mutations? Ie. What is the comparative fitness of these escaped viruses versus the original isolate? This is important information to understand where the virus might evolve to next, or what might be expected if BA7535 was deployed clinically.

7 - BA7535 showed good potency against BA1-5, but was 100-300 fold less active against either ancestral strains or the current XBB and BQ.1 lineages. The authors should provide context in the Discussion about the impacts such a loss of potency would have for the clinical utility of BA7535. Would effective dosing still be possible?

Reviewer #3 (Remarks to the Author):

Overview

The study presented here by Wang et. al. identifies a neutralising epitope targeted by antibody BA7535 that is conserved across SARS-CoV-2 viral variants of concern (VoC). They demonstrate that BA7535 is able to neutralise all previous and current VoC in pseudovirus assays and in authentic SARS-CoV-2 viral neutralisation assays show in vitro neutralisation across Omicron lineage members. Unusually for broadly neutralising antibodies, structural characterisation of BA7535 in complex with Spike reveals an epitope overlapping the ACE-2 receptor binding motif of the RBD. The authors

additionally show in vivo protection against infection (via prophylaxis) in mouse models of Omicron BA.5 infection and therapeutic benefit in a treatment model, particularly when combined with a second broadly neutralising antibody BA7208. The improvement shown in prevention of weight loss in the cocktail therapy versus BA7535 alone (as a treatment) is supported by assays demonstrating the inability of BA7535 to induce efficient ADCC and ADCP.

The data presented here are of high quality and convincingly demonstrate highly potent and broad neutralisation across SARS-CoV-2 lineages by this antibody. It is additionally interesting that this breadth is provided through targeting of the RBM which is extremely variable. Further analysis of the BA7535 binding epitope in the context of ACE2 interaction and particularly related to essential receptor binding residues on the SARS-CoV-2 RBD would help to further determine the reason for this broad neutralisation. This would be highly relevant for future development of targeted antibody therapeutics.

This work will be of interest to the field and represents an update in comprehensive structural and functional characterisation of an antibody capable of broad in vitro neutralisation across current VoC in addition to in vivo neutralisation in a BA.5 mouse challenge model. However, partially characterised BnAbs capable of in vitro neutralisation of all variants up to XBB.1 with moderate to high potency, as well as other Sarbecovirus family members are extant in the literature (e.g DOI: 10.1126/sciadv.ade3470, doi: 10.1016/j.immuni.2023.02.005, BA-2.07 and Omi-42 in 10.1016/j.celrep.2023.112271, doi: 10.1016/j.cell.2022.05.014).

Major Comments

1. Results, Line 234: While I appreciate that the number of antibodies structurally characterised against SARS-CoV-2 is vast. In order to demonstrate the uniqueness and importance of this antibody in terms of its breadth of neutralisation and binding site, I think that the authors may need to broaden their comparisons from benchmarking against only the antibodies licensed for clinical use. For example, comparisons against the other broadly neutralising (up to XBB) antibodies already published (see overview) may help in defining the unique properties of this antibody and epitope.

2. Results, Line 270: Is there a reason why these residues are so conserved? Some work has been done on ACE2 binding requirements for SARS-CoV-2 RBD and which residues are essential or which networks of compensatory mutations may allow continued binding (for example <https://doi.org/10.1371/journal.ppat.1010951>, <https://doi.org/10.1007/s00430-023-00773-w>) could the authors discuss this in the context of their antibody epitope.

Minor Comments

1. Introduction, Line 76 and Fig 1a.: Please include newest VoC XBB.1.16, XBB.1.9, XBB.2.3 and CH.1.1 here. The list of current and emerging VoC are somewhat variable throughout the manuscript and should be unified as far as possible.

2. Introduction, Lines 86-91: While the question asked by the authors here is pertinent, it may need rewording for improved clarity.

3. Figure S4: There is a lot of data presented on each plot here. Could the authors please divide this, perhaps into two plots for each, one of former VoC and one for current/emerging VoC to make the data more easily visible.

4. Figure S5: Has the BLI competition assay been carried out in both directions (i.e BA7535 as antibody A and BA7208 as antibody B) to ensure that the apparent lack of competition is not solely due to displacement of BA7208 by the higher affinity BA7535? Could the authors please include this data.

5. Figures 2D-H: please could the authors make clear the number of replicates and error for each experiment in the figure legend. It is currently unclear if the replicates statement refers to all experiments or only to figure 2H.

6. Table S2: Could the authors please choose a more distinct colour for the light chain H bond interactions as these were difficult to distinguish from the heavy chain at first glance.

7. Figure 4: The current format for figure 4 is not particularly helpful for understanding where these mutations fall within the BA7535 footprint. Could the authors represent the antibody binding site as a footprint on the RBD surface and highlight mutated residues across all lineages on a single figure or series of RBD views?

8. Discussion, Line 350: Should this read "Structural analysis of the Omicron Spike trimer with BA7535-Fab" rather than "Structural analysis of the Omicron Spike trimer with BA7208-Fab"?

9. Discussion, Line 352-354: If the authors state that structural simulation is verified as an efficient method to predict immune escape could they please provide appropriate references here to support this claim?

REVIEWER COMMENTS

Reviewer #1 (Remarks to the Author):

In comparing the binding interface to the other epitope classes, the authors superpose their complex with individual antibody complex structures, but neglect to discuss epitope classes commonly used in the literature:

Classes 1-4 described by Barnes et al Nature 2020 Dec;588(7839):682-687

Class 5 described by Starr et al Nature volume 597, pages 97–102 (2021)

Class 6 described by Rouet et al Nature Communications volume 14, 687 (2023)

The authors should clarify which of these previously described classes their BA7535 (and BA7208) share epitope surface with (given that the cryoEM structure shows overlap with the ACE2 surface, the epitope class is likely to be predominantly 1 or 2). Class 1 generally implies that the RBD must be in an up-conformation to allow binding, but have the authors investigated whether the binding surface might be available with the RBD in the down conformation?

Response: We thank the Reviewer for the valuable suggestion. We reanalyzed the structure of antibody-antigen complexes according to the three publications recommended by the Reviewer. The first publication (Nature 2020 Dec; 588(7839): 682-687) describe epitope class 1-4, briefly,

(1) Class 1 includes neutralizing antibodies encoded by the VH3-53 gene segment with short CDRH3 loops that block ACE2 and bind only to ‘up’ RBDs;

(2) Class 2 includes ACE2-blocking neutralizing antibodies that bind both ‘up’ and ‘down’ RBDs and can contact adjacent RBDs;

(3) Class 3 includes neutralizing antibodies that bind outside the ACE2 site and recognize both ‘up’ and ‘down’ RBDs; and

(4) Class 4 includes previously described antibodies that do not block ACE2 and bind only to ‘up’ RBDs.

The second publication (Nature 2021 Sep; 597(7874): 97-102) described a cryptic antigenic site, and the authors designated it as 'site V' (Class 5). This antibody binding site is facilitated by packing of the heavy chain CDR3 into an RBD crevice at the center of the epitope, together with polar contacts with all three heavy chain CDRs and the light chain CDR2. The surface bound by antibody is constrained by the deleterious effects of mutations on folded RBD expression, and this constraint is likely enhanced by quaternary packing with the NTD in the closed spike trimer.

The third publication (Nature Communications, 2023 Feb 8; 14(1): 687) described a new epitope as Class 6. The authors designated this epitope, spanning a triangular surface between class 5, class 3 and class 1, as class 6. This epitope broadly avoids residues mutated in VOCs, thus demonstrating the broad and effective neutralization of Class 6 antibody. Class 6 antibody is unimpeded and capable of binding to spike in the fully 'down' position.

In this study, we determined the binding site of antibody BA7535, on the top of the RBDs in the spike protein, and all three RBDs are in the 'up' conformation (**revised Fig. 3a**). The revised Figure 3c shows shows BA7535 partially overlaps with the ACE2 epitope. Moreover, we simulated a structure of spike protein with RBD in the 'down' conformation, we found that BA7535 binding site is partially blocked by the adjacent RBD, as shown in revised Figure S8, Therefore, we classify the BA7535 epitope as a Class 1.

In contrast, antibody BA7208 binds to RBD in both 'up' and 'down' conformations (Cell Discov. 2023; 9: 3), and the epitope is not overlap with ACE2 binding site (**revised Fig. S7**). Thus, BA7208 categorized as Class 3.

We have added the above information in the revised manuscript, please see line 221-256.

Revised Fig. 3. Cryo-electron microscopy of Omicron BA.2 Spike protein with BA7535-Fab.

(a) The complex of three BA7535-Fabs (dark purple, Fab heavy-chain; and light purple, Fab light-chain) with Omicron BA.2 Spike Trimer (cyan).

(b) The complex of one BA7535-Fab with Omicron BA.2 Spike RBD, zoomed-in views of BA7535-Fab binding site on Omicron BA.2 RBD, side chain of residues that forms the hydrogen bonds and salt bridge are displayed. H, heavy chain; L, light chain.

(c) the complex of BA.2 RBD and BA7535-Fab is superimposed with complex of RBD and ACE2 (pink).

Revised Fig. S8. The binding site of BA7535 is partially blocked by the adjacent RBD when the RBDs are in the ‘down’ conformation.

(a) Structure of spike trimer with 3 RBDs in the ‘down’ conformation, with one BA7535 Fab displayed binding onto the protomer 1.

(b) Zoomed-in view of BA7535 Fab binding onto the protomer 1 RBD, however, the binding site is partially blocked by the RBD from adjacent protomer 3 RBD (highlighted in red circle).

Revised Fig. S7. BA7535 binding mode analysis compared to other antibodies.

The complex of BA.2 RBD (cyan) and ACE2 (pink) is superimposed with antibodies: (a) BA7535-Fab (purple), (b) LY-Cov555 (yellow), (c) LY-Cov016, (d) A23-58.1 (yellow), (e) REGN10933 (yellow), (f) 2196 (yellow), (g) SA55 (yellow), (h) 2130 (yellow), (i) LY-Cov-1404 (yellow), (j) REGN10987 (yellow), (k) Vir-7831 (yellow), (l) S309 (yellow), (m) SA58 (yellow), and (n) BA7208 (green).

Technical Comments

The authors need to release coordinates in the PDB so that these structures can be properly interrogated by reviewers.

Response: We thank the Reviewer for the suggestion. We have contacted PDB to release our deposited structures to public. The entries 8H7L, 8H7Z, and EMD-34522, EMD-34526 have been scheduled to be released on 2023-08-30.

The figures of the structures are poor: detail in Figure 3b, in particular, is almost uninterpretable (too many light shade pastels; poor resolution preventing meaningful enlargement).

Response: We thank the Reviewer for point out this issue. We have revised the **Fig. 3** followed the Reviewer's advice.

Likewise, Figure 4 is similarly difficult to interpret due to low resolution. Figure 4 would be improved by only showing a handful of high-resolution images that explicitly highlight the point mutants in close proximity to the antibody interface.

Response: We thank the Reviewer for the suggestion. We have reduced the content in the original **Fig. 4** and enlarged the images to display the mutation sites more clearly. There are 20 mutations from 42 Omicron sub-lineages, 3 mutations that may interfere antibody BA7535 binding are shown in red in revised **Fig. 4a**, and other mutations that not located in the BA7535 epitope are shown in blue in revised **Fig. 4b**.

Revised Fig. 4. BA7535 binding site avoids most mutations from Omicron sub-lineages. The complexes between BA7535-Fab (purple) and RBD of Omicron sub-lineages (cyan). **(a)** Three mutation residues which are located close to the BA7535 binding site are highlighted in red. **(b)** Mutation residues that are not located to the BA7535 binding site are highlighted in blue.

The structure they present is one derived from cryo-EM, although the nominal resolution at the RBD-Fab interface is poor (4 Å), optimised to only about 3 Å. It worries me that they then plug this into PISA from which they derive all sorts of metrics – buried surface, deltaG and hydrogen bonds etc – which is ordinarily fine, but it is a red flag to me that they list a bunch of hydrogen bonds with distances that to my mind are too large to place any confidence in, especially given the low resolution of the model. To my mind, if the distance is larger than 3.2 Å, it's too far to be confidently ascribed as an H-bond, yet distances of 3.4, 3.5, 3.6 and 3.6 are listed. Additionally, in Table S3 they list buried surface to 4 significant figures (two or three would be more realistic given that the number can vary +/- 50 Å² by changing rotamer of a single residue). This is symptomatic of what I would call general over interpretation of low-resolution structural data and extends to the generation of in silico point mutants: it wasn't clear how these were derived and what rotamers were chosen.

Response: We agree with the Reviewer that a low resolution of a protein structure will limit the accuracy of the prediction of structure-based properties such as hydrogen bond, salt-bridge, interface area, and solvation free energy. Particularly for hydrogen bond, the predicted donor-acceptor geometry would bear certain error unless the hydrogen atoms can be accurately placed, which unfortunately is unavailable for most protein structures determined experimentally. However, given a refined structure, useful insights can still be obtained by looking at those structural properties, as exemplified in recent studies using PISA (Sci Rep. 2023, 13(1): 13496.; Elife. 2023, 12: RP86784). In PISA, the criteria as illustrated in the **Response Fig. 1**, which used to identify possible hydrogen bonds (CompLife 2005 pp 163–174; J Mol Biol. 2007 372(3): 774-97; Prog Biophys Mol Biol. 1984, 44(2): 97-179), where the maximum O-O(N) distance was taken as 3.5 Å. Although fairly generous angle criteria for the acceptance of hydrogen bond were used as the algorithm tried to include all reasonable hydrogen bonds suggested by the structural data (including weak bifurcated hydrogen bonds, Prog Biophys Mol Biol. 1984, 44(2): 97-179.), we would like to clarify that the O-O(N) distance criterion of 3.5 Å in H-bond determination is rational and have been widely used in the computational biophysics community. Therefore, we suppose that the distances listed by PISA are reasonable to give possible hydrogen bonds. Regarding the report of the interface area numbers, we agree with the reviewer and have revised Table S3 and Table S5 to keep only 3 significant figures. All mutation systems were generated by building the side-chain atoms in the most frequently occurring rotamer using Coot (Acta Crystallogr D Biol Crystallogr. 2010, 66(Pt 4): 486-501.). We have also carefully reexamined the wording of the text to avoid any overinterpretation of the results.

[redacted]

Response Fig. 1. Geometrical criteria used for defining hydrogen bonds. Where hydrogen positions were available, O-H(maximum) = 2.5 Å. Where no hydrogen positions were available O-O(N)(maximum) = 3.5 Å. Angle at the oxygen atom 90-180°; angle at the hydrogen atom 90-180°, figure adapted from Prog Biophys Mol Biol. 1984, 44(2): 97-179.

Reviewer #2 (Remarks to the Author):

The manuscript by Wang and colleagues details a human monoclonal antibody (BA7535) with broad neutralisation activity against SARS-CoV-2 variants. Using a combination of vaccination of humanised mice, followed by phage display, the authors recovered BA7535 and confirmed broad neutralisation activity in vitro and in vivo. Potency was strongest against BA1-BA5 lineages, with significant loss of activity against ancestral and more recent strains. Structural analysis reveals an epitope at the top of the RBD and bioavailable in the “up” conformation. The paper is clearly presented, although the information gain from a single antibody epitope at this stage of the pandemic is somewhat limited.

Comments:

1 – This antibody was elicited by immunisation of a humanised mouse then phage display, which may limit applicability to human responses to SARS-CoV-2. Can the authors identify any human antibodies that bind with similar parameters (or share genetic features) in the extensive BCR sequencing or mAb datasets available? i.e. how common are such antibodies?

Response: We thank the reviewer for this valuable suggestion. We have analyzed the genetic feature of mAb BA7535 and revealed that BA7535 was derived from the pairing of IGHV3-30 and IGKV5-2. We also analyzed the germline usage of 5712 SARS-CoV-2 spike-specific human mAbs from the COV-AbDab database (doi: 10.1093/bioinformatics/btaa739.). This result shows that the BA7535 class of antibodies with IGHV3-30/IGKV5-2 pairing is rare in the human antibodies, suggesting that this class of antibodies dose not appear to be induced preferentially in human population by SARS-

CoV-2 infection or vaccination (**Fig. S14**). This is one of the advantages of using humanized mouse immunization and phage display technology, which allows us to screen for broad neutralizing antibodies that are rare in humans. Corresponding revision was performed in line 428-436.

Revised Fig. S14 Germline usage analysis of 5712 SARS-CoV-2 spike-specific human mAbs from the COV-AbDab database. BA7535 is derived from the pairing of IGHV3-30-3 and IGKV5-2.

2- The ADCC and ADCP results are confounding given most spike-specific mAbs should give some signal in this assay. The authors should provide a rationale in the Discussion for why BA7535 and LY-CoV1404 fail to engage FcR.

Response: it is an enlightening comment. Indeed, we are now conducting another study on the relationship between ADCC, ADCP function and the antibody targeted epitope.

Antibodies can elicit a number of mechanisms to delete target cells, including antibody dependent cellular cytotoxicity (ADCC) and antibody dependent cellular phagocytosis (ADCP). Fab and Fc function may even be considered interdependent. The inherent properties of the target epitope help define which of these mechanisms are more important for efficacy. However, why mAb binding to different epitopes within the same target elicits different levels of therapeutic activity (ADCC and ADCP), is often unclear. Some studies (ref 1) show that CDC and ADCC favored a membrane proximal epitope, whilst ADCP favored an epitope positioned further away. Altering the position of the antibody epitope is able to change the effector mechanisms engaged and facilitates the selection of mAbs designed to delete target cells. In addition, some influenza virus research also indicated that antibody affinity and epitope accessibility influence competition-mediated ADCC inhibition (ref 2), and ADCC Inhibition by HAI⁺ antibodies is achieved through competitive binding to HA on virus particles and on the surface of infected cells. Besides, Antibodies that predominantly bind monovalently (such as those with moderate affinity or those engineered for monovalent binding) may elicit stronger effector functions because higher cell surface densities of Fc domains can be achieved (ref 3). In our recent study results indicated that the RBD antibodies that possess ADCC and ADCP function tend to bind to the RBD3, RBD4 and RBD6 epitope. The epitope, specificity, affinity and antibody concentration all affect antibody functional response activation. According to the advice we added the corresponding discussion in line 422-424.

Reference

1. Cleary KLS, Chan HTC, James S, et al., Antibody Distance from the Cell Membrane Regulates Antibody Effector Mechanisms. *J Immunol.* 2017 May 15;198(10):3999-4011.
2. He W, Tan GS, Mullarkey CE, et al., Epitope specificity plays a critical role in regulating antibody-dependent cell-mediated cytotoxicity against influenza A virus. *Proc Natl Acad Sci U S A.* 2016 Oct 18;113(42):11931-11936.
3. Oostindie, S.C., Lazar, G.A., Schuurman, J. et al. Avidity in antibody effector functions and biotherapeutic drug design. *Nat Rev Drug Discov* 21, 715–735 (2022).

3 – Line 230 -” the risk of being affected by mutation of RBD sites and enhance the broad spectrum while improving the neutralization efficiency. In vitro neutralizing potency experiments also confirmed the broad spectrum of BA7535 (Fig. 1a). In conclusion, among these antibodies mentioned above, BA7535 is an efficient one with broad neutralization potency”

What is meant by efficiency in this case? BA7535 neutralises with comparable or lesser potency than many of the comparator mAbs named.

Response: Thanks for your correction, we revised the statement to avoid the ambiguity in line 252-253.

4- Line270 - “In summary, antibody BA7535 has a broad spectrum against all Omicron variants.” This is overly strong and not confirmed by the data but predicted by the authors based on surface areas not experimental validation. The authors too concede that local conformational changes can affect hydrogen bonding between the epitope and paratope, as such this language should be more circumspect.

Response: We thank the reviewer for this advice. We revised the statement in line 291-292 and 254-256.

5 – Somewhat confoundingly, BA7535 treatment (either 10mg/kg or 2mg/kg) showed no protection against weight loss despite eliminating viral replication and spread. How do the authors account for this? Animals terminated by day 4, so it is difficult to ascertain a complete picture of pathogenesis. This study should be repeated with completed weight analysis until recovery or humane endpoint.

Response: As the reviewer’s advice, we repeated the animal experiment and monitored the weight loss and survival rate until endpoint. Corresponding revision was added in line 310-320. Because the K18-hACE2 model resulted in more severe disease, manifesting in weight loss, and replication in multiple organs – including lung, brain, and gut. In this study treatment with neutralizing antibody failed to prevent weight loss and death in vivo challenged with high dose of SARS-CoV-2 Omicron BA.5 (1×10^5 FFU) in agreement with previous reports (ref1, ref 2), which suggest that neutralizing antibodies can be

overwhelmed by a sufficiently high virus inoculum. But the lung viral titer, alleviated lung injury and longer survival duration confirmed the protection of BA7535 in vivo.

The K18 hACE2 mouse used in this study was pursued from Gem Pharmatech Co., Ltd. It's worth noting that, the H11-K18-hACE2 mouse from Gem Pharmatech Co., Ltd is different from that of the Jackson Laboratory. The K18-hACE2 mouse from Gem Pharmatech is a lethal model, even at low dose (ref 3,4). As for Jackson mouse, inoculation with higher viral doses (2×10^3 and 2×10^4 PFU) of SARS-CoV-2 caused lethality of all mice and severe damage of various organs, including lung, liver, and kidney, while lower doses (2×10^1 and 2×10^2 PFU) led to less severe tissue damage and some mice recovered from the infection (ref 5). Both animal model infected with SARS-CoV-2 will caused multi-organ injury or failure.

Reference

1. Ng KW, et al. SARS-CoV-2 S2-targeted vaccination elicits broadly neutralizing antibodies. *Science translational medicine* 14, eabn3715 (2022).
2. Zhou X, et al. A novel hACE2 knock-in mouse model recapitulates pulmonary and intestinal SARS-CoV-2 infection. *Frontiers in microbiology* 14, 1175188 (2023).
3. Bai, L., Zhao, Y., Dong, J. et al. Coinfection with influenza A virus enhances SARS-CoV-2 infectivity. *Cell Res* 31, 395–403 (2021).
4. Zhao, S., Zhang, H., Yang, X. et al. Identification of potent human neutralizing antibodies against SARS-CoV-2 implications for development of therapeutics and prophylactics. *Nat Commun* 12, 4887 (2021).
5. Dong W, Mead H, Tian L. et al. The K18-Human ACE2 Transgenic Mouse Model Recapitulates Non-severe and Severe COVID-19 in Response to an Infectious Dose of the SARS-CoV-2 Virus. *J Virol.* 2022 Jan 12;96(1):e0096421. doi: 10.1128/JVI.00964-21. Epub 2021 Oct 20. PMID: 34668775; PMCID: PMC8754221.
- 6- It is interesting the viruses so rapidly escaped (by P3-4) under antibody selection pressure. Are there any fitness cost associated with these mutations? Ie. What is the comparative fitness of these escaped viruses versus the original isolate? This is important information to understand where the virus might evolve to next, or what might be expected if BA7535 was deployed clinically.

Response: We thank the reviewer for this advice. We provided corresponding discussion as below: “As for the generation of SARS-CoV-2 escape mutations against BA7535 during serial passage, the escape mutant N455K emerged in the presence of BA7535 after the third passage and became readily fixed in the population by the fourth passage. But the IC₅₀ of BA7535 only increased from 2.8 ng/mL to 14.6 ng/mL and still performed potent neutralizing activity, the mutation N455K did not contribute to the immune escape sharply. Apart from that, serial passage did not generate escape mutants in the presence of BA7535/BA7208 cocktail, which clear the way for clinical application of BA7535. We also detected the the binding affinity of BA7535 to BA.5 RBD and mutant RBD (N455K), to some extent, there was mild decrease in binding affinity of mutant RBD (N455K) (KD=1.625E-8) against ACE2 compared with wild type RBD(KD=5.586E-9), which may influence the fitness (**Fig. S13**).”. The corresponding revision was added in line 403-410

Fig. S13. Comparison of binding affinity between BA7535 VS BA.5 RBD455N and BA7535 VS BA.5 RBD455K

7 - BA7535 showed good potency against BA1-5, but was 100-300 fold less active against either ancestral strains or the current XBB and BQ.1 lineages. The authors should provide context in the Discussion about the impacts such a loss of potency would have for the clinical utility of BA7535. Would effective dosing still be possible?

Response: We thank the reviewer for this advice. We added corresponding discussion as below: “BA7535 exhibited exceptional breadth and was able to neutralize all tested variants. No one variant tested was observed to escape from neutralization. Though SARS-CoV-2 wild type(wt) and D614G strains only impaired the potency of BA7535, the combination of BA7535 and BA7208 overcome immune escape and provided broader neutralization potency against SARS-CoV-2 wt and D614G variant than alone.” The

corresponding revision was added in line 398-402. In addition, BA7208 still performed efficient protection against SARS-CoV-2 wild type and D614G in mouse model and according to our experience BA7535 with low level of neutralizing activity (IC50: 0.6 ug/ml) can still performed protection against SARS-CoV-2 in vivo (ref 1,2).

Reference

1. Wang Y, Yan A, Song D, et al., Biparatopic antibody BA7208/7125 effectively neutralizes SARS-CoV-2 variants including Omicron BA.1-BA.5. *Cell Discov.* 2023 Jan 7;9(1):3.
2. Song D, Wang W, Dong C, et al. Structure and function analysis of a potent human neutralizing antibody CA521FALA against SARS-CoV-2. *Commun Biol.* 2021 Apr 23;4(1):500.

Reviewer #3 (Remarks to the Author):

Overview

The study presented here by Wang et. al. identifies a neutralising epitope targeted by antibody BA7535 that is conserved across SARS-CoV-2 viral variants of concern (VoC). They demonstrate that BA7535 is able to neutralise all previous and current VoC in pseudovirus assays and in authentic SARS-CoV-2 viral neutralisation assays show in vitro neutralisation across Omicron lineage members. Unusually for broadly neutralising antibodies, structural characterisation of BA7535 in complex with Spike reveals an epitope overlapping the ACE-2 receptor binding motif of the RBD. The authors additionally show in vivo protection against infection (via prophylaxis) in mouse models of Omicron BA.5 infection and therapeutic benefit in a treatment model, particularly when combined with a second broadly neutralising antibody BA7208. The improvement shown in prevention of weight loss in the cocktail therapy versus BA7535 alone (as a treatment) is supported by assays demonstrating the inability of BA7535 to induce efficient ADCC and ADCP.

The data presented here are of high quality and convincingly demonstrate highly potent and broad neutralisation across SARS-CoV-2 lineages by this antibody. It is additionally interesting that this breadth is provided through targeting of the RBM which is extremely

variable. Further analysis of the BA7535 binding epitope in the context of ACE2 interaction and particularly related to essential receptor binding residues on the SARS-CoV-2 RBD would help to further determine the reason for this broad neutralisation. This would be highly relevant for future development of targeted antibody therapeutics.

This work will be of interest to the field and represents an update in comprehensive structural and functional characterisation of an antibody capable of broad in vitro neutralisation across current VoC in addition to in vivo neutralisation in a BA.5 mouse challenge model. However, partially characterised BnAbs capable of in vitro neutralisation of all variants up to XBB.1 with moderate to high potency, as well as other Sarbecovirus family members are extant in the literature (e.g DOI: 10.1126/sciadv.ade3470, doi: 10.1016/j.immuni.2023.02.005, BA-2.07 and Omi-42 in 10.1016/j.celrep.2023.112271, doi: 10.1016/j.cell.2022.05.014).

Major Comments

1. Results, Line 234: While I appreciate that the number of antibodies structurally characterised against SARS-CoV-2 is vast. In order to demonstrate the uniqueness and importance of this antibody in terms of its breadth of neutralisation and binding site, I think that the authors may need to broaden their comparisons from benchmarking against only the antibodies licensed for clinical use. For example, comparisons against the other broadly neutralising (up to XBB) antibodies already published (see overview) may help in defining the unique properties of this antibody and epitope.

Response: We thank the Reviewer for the suggestion and literature recommendation, we reanalyzed the structure of antibody-antigen complexes according to the three publications recommended by the Reviewer.

we have read these publications. In the first publication (Sci Adv. 2023, 9(30): eade3470, DOI: 10.1126/sciadv.ade3470), Chia et al. reported the isolation and characterization of highly potent mAbs targeting the RBD. Among the six mAbs identified, one (E7) showed better huACE2-dependent sarbecovirus neutralizing potency and breadth than any other mAbs reported to date. Mutagenesis and cryo-electron microscopy studies indicate E7 binds to a quaternary epitope across an 'up' RBD and a neighboring 'down' RBD. Most of its interaction is on the 'up' RBD, a large part of the E7 epitope overlaps with the Class

1 mAbs epitopes, while the remaining parts overlap with Class 4 mAb epitopes. On the 'down' RBD, E7 binds to two residues that are located within the previously described Class 2 epitope. We have created a new **Fig. S9** to compare the antibody binding mode compare to BA7535.

In the second publication (Immunity, 2023, 56(3): 669-686.e7, doi: 10.1016/j.immuni.2023.02.005) Zhou et al. reported an isolation of a large panel of broadly neutralizing antibodies (bnAbs) from SARS-CoV-2 recovered-vaccinated donors, which targets a conserved S2 region. Structural studies (X-ray crystallization, rather than cryo-EM, to determine the structures of antibodies and stem-helix peptides complexes) of these bnAbs delineated the molecular basis for their broad reactivity. They found out the epitope site is located at the interface within a coiled-coil helix bundle at the base of the prefusion SARS-CoV-2 spike and is connected to heptad repeat region 2 (HR2). Both helix and heptad repeat undergo dramatic conformational changes between prefusion and postfusion states. Antibody binding to this stem-helix epitope site may block this transition and hence membrane fusion. Because the structures are antibodies with peptides, nor RDB or spike protein, we didn't include these antibodies in the **Fig. S9**.

In the third publication (Cell Rep, 2023, 42(4): 112271, 10.1016/j.celrep.2023.112271), Djokaite-Guraliuc et al. described a panel of 25 potent monoclonal antibodies (mAbs) generated from vaccinees suffering BA.2 breakthrough infections. Epitope mapping shows potent mAb binding are clustered into the three regions closest to the ACE2-binding site: the right shoulder, the neck, and the left shoulder. They determined the cryo-EM structure of the complex of the Delta-RBD with the antibodies. BA.2-23 is an IGHV3-53 family antibody and binds the left shoulder in the expected position. BA.2-10 (IGHV3-9) binds at the right chest region of the RBD. It makes extensive interactions via complementarity-determining regions (CDRs) H3, L3, and L1 with residue V445 of the RBD and with R346 via CDRs H1 and H2. BA.2-13 (IGHV3-15) also binds to the right chest region but higher and more toward the midline of the RBD than BA.2-10 and in a very different orientation so that its LC contact area on the RBD largely overlaps with the footprint of BA.2-10 HC. BA.2-36 (IGHV4-61) binds to the right chest region in a similar

position and orientation to BA.2-13 We have created a new **Fig. S9** to compare the antibody binding mode of BA.2-23, BA.2-36, BA.2-10 and BA.2-13.

In the forth publication (Cell, 2022, 185(12): 2116-2131.e18, doi: 10.1016/j.cell.2022.05.014) Nutalai et al. analyzed structure-and-function of 27 potent RBD-binding mAbs isolated from vaccinated volunteers following breakthrough Omicron-BA.1 infection reveals that they are focused in two main clusters within the RBD, with potent right-shoulder antibodies showing increased prevalence. The authors provided structural information on the 11 potent antibodies use either cryo-EM or X-ray crystallography approaches. We included 6 antibodies that the authors claimed were good on all SARS-CoV-2 variants of concern in the new **Fig. S9**, to compare the binding sites to BA7535.

According to the latest representative antibodies, we added the above information and made comparison in the revised manuscript, please see line 221-250.

Fig. S9. BA7535 binding mode analysis compared to other antibodies. The complex of BA.2 RBD (cyan) and ACE2 (pink) is superimposed with antibodies: (a) BA7535-Fab (purple), (b) E7 (yellow), (c) Omi-2(yellow), (d) Omi-12 (yellow), (e) Omi-3 (yellow), (f) Omi-18 (yellow), (g) Omi-25 (yellow), (h) Omi-42 (yellow), (i) BA.2-23 (yellow), (j) BA.2-10 (yellow), (k) BA.2-13 (yellow), (l) BA.2-36 (yellow).

2. Results, Line 270: Is there a reason why these residues are so conserved? Some work has been done on ACE2 binding requirements for SARS-CoV-2 RBD and which residues are essential or which networks of compensatory mutations may allow continued binding (for

example, <https://doi.org/10.1371/journal.ppat.1010951>, <https://doi.org/10.1007/s00430-023-00773-w>) could the authors discuss this in the context of their antibody epitope.

Response: We thank the Reviewer for point out this issue. We read these publications and added some discussion as below.

In the first publication (PLoS Pathog, 2022, 18(11): e1010951, doi.org/10.1371/journal.ppat.1010951), Starr et al. determined the impacts of all single amino acid mutations in the Omicron BA.1 and BA.2 RBDs on ACE2-binding affinity, RBD folding, and escape from binding by the antibody LY-CoV1404. They found out that Omicron variants show additional lineage-specific shifts, including examples of the epistatic phenomenon of entrenchment that causes the Q498R and N501Y substitutions present in Omicron to be more favorable in that background than in earlier viral strains. They find that the Omicron BA.1 and BA.2 RBDs are highly tolerant to mutation and can increase affinity for ACE2. In the case of Omicron, some of these affinity-enhancing mutations are reversions or secondary changes at sites that mutated during Omicron's emergence (e.g., mutations of residues N417 or R493). Several substitutions in Omicron such as N501Y and Q498R exhibit epistatic entrenchment, where the derived state is more favorable for ACE2 binding. In contrast, the Q493R substitution does not show any evidence of entrenchment. Instead of becoming more tolerable in combination with the other Omicron mutations, as would occur for entrenchment, R493 is more unfavorable for ACE2 binding in the BA.1 and BA.2 backgrounds. Last, they determined how mutations in each RBD background facilitate escape from binding by LY-CoV1404. Their data

suggest that LY-CoV1404 may have lower baseline affinity for BA.1 and BA.2 RBDs that opens up these additional pathways of escape.

In the second publication (Med Microbiol Immunol, 2023, 212(4): 291-305, doi.org/10.1007/s00430-023-00773-w), Haycroft et al. comprehensively investigated the impact of RBD mutations, including 5 variants of concern or interest - including Omicron (BA.2) - and 33 common point mutations, both on IgG recognition and ACE2-binding inhibition. The RBD is a mutational hotspot for SARS-CoV-2. Because of its position at the ACE2 interface, mutations in the RBD can alter ACE2 affinity as well as result in the loss of epitopes for antibodies. Their data highlight the influence of particular point mutations on recognition of the RBD by IgG antibodies and naturally occurring RBD mutations on the potential for polyclonal antibodies to recognize and block ACE2 binding. Then authors have shown single-point mutations in RBD (such as N501Y found in Alpha, Beta, Gamma and Omicron) can enhance affinity to ACE2 and their results suggest that beyond antibody recognition, affinity interactions between RBD and ACE2 exert another layer of influence on the potential for antibodies to effectively block ACE2-binding and achieve neutralization.

All above finding indicated that the epidemic mutations always influence the virus infectivity or fitness. We added corresponding discussion in line 386-397 as below: "Structure analysis indicated that the epitope targeted by BA7535 was highly conserved among SARS-CoV-2 variants. It is difficult to predict whether the conserved epitope-targeted antibody or vaccine will be escaped in future. Mutation probability of conserved epitope was determined by many factors, including virus structural stability, infectivity, spike-ACE2 binding affinity, host response and selective pressure. Some conserved residues are essential to ACE2 binding or structural stability. As SARS-CoV-2 uses RBD to interact with host angiotensin-converting enzyme 2 (ACE2) and ensure cell recognition. The epidemic mutations always influence the virus infectivity or fitness. Meanwhile, most variants convergently acquired similar amino acid substitutions at critical residues in the spike. Epidemic dynamics modelling also suggests that most hot mutations or substitutions increase viral fitness. All above indicated that it is difficult to mutate for the conserved epitope involving viral structure or stability".

Minor Comments

1. Introduction, Line 76 and Fig 1a.: Please include newest VoC XBB.1.16, XBB.1.9, XBB.2.3 and CH.1.1 here. The list of current and emerging VoC are somewhat variable throughout the manuscript and should be unified as far as possible.

Response: As the Reviewer's advice, we added the newest VoCs in the statement (line 106-111) and Fig 1a, Fig 2a, Fig 4 and Table S4.

2. Introduction, Lines 86-91: While the question asked by the authors here is pertinent, it may need rewording for improved clarity.

Response: As the Reviewer's advice, we reworded the statement in line 87-92.

3. Figure S4: There is a lot of data presented on each plot here. Could the authors please divide this, perhaps into two plots for each, one of former VoC and one for current/emerging VoC to make the data more easily visible.

Response: As the Reviewer's advice, we divided each plot into two parts, one of previous VoCs, and one is for the current/emerging VoCs in **Fig. S4**.

4. Figure S5: Has the BLI competition assay been carried out in both directions (i.e BA7535 as antibody A and BA7208 as antibody B) to ensure that the apparent lack of competition is not solely due to displacement of BA7208 by the higher affinity BA7535? Could the authors please include this data.

Response: We thank the Reviewer for the suggestion. We performed additional BLI competition assay in both directions and updated the **Fig. S5**.

5. Figures 2D-H: please could the authors make clear the number of replicates and error for each experiment in the figure legend. It is currently unclear if the replicates statement refers to all experiments or only to figure 2H.

Response: We added the number of replicates and error for each experiment in the **Fig. 2** legend.

6. Table S2: Could the authors please choose a more distinct colour for the light chain H bond interactions as these were difficult to distinguish from the heavy chain at first glance.

Response: We thank the Reviewer for the suggestion. We have revised the **Table S2** to highlight the residues in the spike RBD and antibody light chain and heavy chain, the colors are consistent with **Fig. 3b**.

7. Figure 4: The current format for figure 4 is not particularly helpful for understanding where these mutations fall within the BA7535 footprint. Could the authors represent the antibody binding site as a footprint on the RBD surface and highlight mutated residues across all lineages on a single figure or series of RBD views?

Response: We thank the Reviewer for the suggestion. We have redrawn Figure 4 to display the antibody binding site as a footprint on the RBD surface. There are 20 mutations from 42 Omicron sub-lineages, 3 mutations that may interfere antibody BA7535 binding are shown in red in revised **Fig. 4a**, and other mutations that not located in the BA7535 epitope are shown in blue in revised **Fig. 4b**.

8. Discussion, Line 350: Should this read "Structural analysis of the Omicron Spike trimer with BA7535-Fab" rather than "Structural analysis of the Omicron Spike trimer with BA7208-Fab"?

Response: We thank the Reviewer for pointing out this typo. We have corrected it in the revised manuscript, please see page 15 line 377-378.

9. Discussion, Line 352-354: If the authors state that structural simulation is verified as an efficient method to predict immune escape could they please provide appropriate references here to support this claim?

Response: We thank the Reviewer for the suggestion. We have cited the publication by Cao et al., Nature, 2022, 602: 657-663 in the revised manuscript, please see page 15 line 379-380.

REVIEWER COMMENTS

Reviewer #1 (Remarks to the Author):

The reviewers have addressed all comments

Reviewer #2 (Remarks to the Author):

The authors have addressed my major concerns and updated the manuscript accordingly.

Reviewer #3 (Remarks to the Author):

Here the authors present a thorough characterisation of their broadly neutralising antibody BA7535 with a more complete functional and structural examination of their candidate antibody than is found in many SARS-CoV-2 mAb publications, which would provide guidance to the field in terms of future avenues for SARS-CoV-2 mAb therapeutics.

While the authors have gone to considerable effort to address the reviewers questions and have markedly improved the paper, there is still a lack of detail in key areas, particularly in relating their antibody to those currently in the literature. This is particularly relevant as the premise of this manuscript is the identification of a novel conserved neutralising epitope that enables broad spectrum neutralisation of SARS-CoV-2 variants.

The additional figure S9 fails to show whether antibody epitopes overlap that of BA7535 or give any details of unique contact residues for BA7535 and the modified results section (lines 246-256) accompanying the figure is unclear. I am therefore unable to determine from the modified paper whether the epitope targeted by BA7535 is indeed novel and if not, what distinguishes this antibody from other broadly neutralising antibodies targeting similar epitopes.

REVIEWER COMMENTS

Reviewer #1 (Remarks to the Author):

The reviewers have addressed all comments

Response: thank you

Reviewer #2 (Remarks to the Author):

The authors have addressed my major concerns and updated the manuscript accordingly.

Response: thank you

Reviewer #3 (Remarks to the Author):

Here the authors present a thorough characterisation of their broadly neutralising antibody BA7535 with a more complete functional and structural examination of their candidate antibody than is found in many SARS-CoV-2 mAb publications, which would provide guidance to the field in terms of future avenues for SARS-CoV-2 mAb therapeutics.

While the authors have gone to considerable effort to address the reviewers questions and have markedly improved the paper, there is still a lack of detail in key areas, particularly in relating their antibody to those currently in the literature. This is particularly relevant as the premise of this manuscript is the identification of a novel conserved neutralising epitope that enables broad spectrum neutralisation of SARS-CoV-2 variants. The additional figure S9 fails to show whether antibody epitopes overlap that of BA7535 or give any details of unique contact residues for BA7535 and the modified results section (lines 246-256) accompanying the figure is unclear. I am therefore unable to determine from the modified paper whether the epitope targeted by BA7535 is indeed novel and if not, what distinguishes this antibody from other broadly neutralising antibodies targeting similar epitopes.

Response: We thank the Reviewer for the valuable suggestion. We reanalyzed the structure of antibody-antigen complexes and made comparison of BA7535 interacting residues with previous reported antibodies.

Firstly, we revised Figure S9 and show the detailed epitopes targeted by BA7535 and revised the statements as below:

In addition to the antibodies that are licensed for clinical use, there are several reports that characterized the latest-developed antibodies binding to spike or RBD by cryo-EM

and X-ray crystallography^{38, 39, 40, 41}. we also compared the binding site of BA7535 with these antibodies, and Supplementary Fig. 9 illustrated that most newly developed antibody epitopes are located on the top of RBD, similar with BA7535, block the ACE2 binding site. Among these antibodies, BA7535, Omi-3, Omi-2, Omi-12, Omi-25, and BA.2-36 have their epitopes that directly overlap with ACE2 binding site, and more important, BA7535 and Omi-25 share unique contact residues on the top of the RBD. Based on the structural characteristics, BA7535-Fab bind to the top portion thus avoids most of the mutational sites in RBD (Fig. 4), it reduces the risk of being affected by most mutations in spike protein and enhance the broad spectrum.(line 248-255)

Supplementary Figure. 9. BA7535 binding mode analysis compared to other antibodies.

The complex of BA.2 RBD (cyan) and ACE2 (pink) is superimposed with antibodies displayed in as ribbon, and in the zoomed-views, the binding site of ACE2 (pink) and epitopes of each antibody (purple for BA7535 and yellow for other antibodies) are highlighted on the surface of RBD structure (cyan), with the epitopes that overlap with ACE2 binding site shown in red. (a) BA7535-Fab (purple), (b) E7 (yellow), (c) Omi-2(yellow), (d) Omi-12 (yellow), (e) Omi-3 (yellow), (f) Omi-18 (yellow), (g) Omi-25 (yellow), (h) Omi-42 (yellow), (i) BA.2-23 (yellow), (j) BA.2-10 (yellow), (k) BA.2-13 (yellow), (l) BA.2-36 (yellow).

Secondly, we downloaded 221 published RBD-specific mAbs with available structures, Comparative analysis of epitopes of BA7535 and 221 published RBD-specific mAbs indicated that most of previously reported neutralizing antibodies target one or two major antigenic sites, single or double mutations of concern at key positions always lead to immune escape, while the epitope residues and buried surface area (BSA) for each epitope residue of BA7535 are more dispersive and appear more resistant to most mutants. In vitro neutralizing potency experiments also confirmed the broad spectrum of BA7535. Detailed revision was provided as below (line 258-270):

Comparative analysis of epitopes of BA7535 and published RBD-specific mAbs

A total of 221 human SARS-CoV-2 RBD-specific mAbs with available structures were downloaded from the PDB (<https://www.rcsb.org/>). The PDB accession codes for the 221 mAbs were summarized in supplementary Table X. Epitope residues and buried surface area (BSA) for each epitope residue of the 221 RBD-specific mAbs as well as BA7535 were determined using the PDBePISA server (https://www.ebi.ac.uk/msd-srv/prot_int/). BSA for each epitope residue is considered a feature of a certain antibody and used to construct a feature matrix $MA \times B$ for downstream analysis, where A is the number of antibodies and B is the number of features (amino acid length of RBD: V320-K529). Therefore, a BSA matrix with 221 rows and 210 columns for the 221 RBD mAbs was obtained, which were subsequently used as inputs for epitope classification with the R

package UMAP (v 0.2.9.0). After the dimensionality reduction analysis, the 221 mAbs were clustered into seven groups, namely RBD I-VII. The BSA for each epitope residue of the 221 RBD-specific mAbs as well as BA7535 were shown as heatmap (supplementary Figure X) using the R package ComplexHeatmap (v 2.14.0).

Supplementary Figure. 10. Comparative analysis of epitopes of BA7535 and 221 published RBD-specific mAbs.

Thirdly, the epitope in the Omicron RBD for BA7535 was also marked in manuscript as below: The epitope in the Omicron RBD for BA7535 is composed of 7 interacting residues, including T415, D420, Y421, A475, N487, Y489 and R493. Fig. 3b reveals six hydrogen bonds and one salt bridge formed between residues from BA7535-Fab and residues from Omicron BA.2-RBD (Supplementary Table 2). The hydrogen bonds include T415-Y106, D420-Y106, Y421-L103, A475-T28, N487-R98 and Y489-R98 (fore residues are from Omicron BA.2 RBD and hind residues are from BA7535-Fab) and the salt bridge is formed between R493-E50 (line 209-215).

Fig. 3 Cryo-electron microscopy of Omicron BA.2 Spike protein with BA7535-Fab.

(a) The complex of three BA7535-Fabs (dark purple, Fab heavy-chain; and light purple, Fab light-chain) with Omicron BA.2 Spike Trimer (cyan).

(b) The complex of one BA7535-Fab with Omicron BA.2 Spike RBD, zoomed-in views of BA7535-Fab binding site on Omicron BA.2 RBD, side chain of residues that forms the hydrogen bonds and salt bridge are displayed. H, heavy chain; L, light chain.

(c) the complex of BA.2 RBD and BA7535-Fab is superimposed with complex of RBD and ACE2 (pink).

REVIEWERS' COMMENTS

Reviewer #3 (Remarks to the Author):

I thank the authors for their responses to my comments. They have addressed all my concerns with the detailed and comprehensive additional analyses performed here.

REVIEWER COMMENTS

Reviewer #3:

I thank the authors for their responses to my comments. They have addressed all my concerns with the detailed and comprehensive additional analyses performed here.

Response: thank you